## [Peer Review File · Nature Immunology]

Metabolic quiescence of naïve-like memory T cells precedes and maintains antigen-specific T-cell memory

Corresponding Author: Professor Kilian Schober

Version 0:

Decision Letter:

16th Jun 2025

Dear Dr. Schober,

Thank you for your response to the referees' comments on your Article, "Metabolic quiescence precedes and maintains human antigen-specific T-cell memory in vivo". While we find your work of potential interest, the reviewers have raised substantial concerns that must be addressed. As such, we cannot accept the current version of the manuscript for publication, but would be happy to consider a revised version that addresses these concerns, as long as novelty is not compromised in the interim.

Please revise the manuscript to address all issues raised by the referees and according to your letter. At resubmission, please include a point-by-point "Response to referees" detailing how you have addressed each referee comment (please specify the page and figure number where the new data can be found in the revised manuscript and please highlight the changes in the manuscript as well). This response will be sent back to the referees along with the revised manuscript.

In addition, please include a revised version of any required reporting checklist. It will be available to referees (and, potentially, statisticians) to aid in their evaluation if the manuscript goes back for peer review. A revised checklist is essential for re-review of the paper.

The Reporting Summary can be found here:
<https://www.nature.com/documents/nr-reporting-summary.pdf>

Extended Data figures and tables are online-only (appearing in the online PDF and full-text HTML version of the paper), peer-reviewed display items that provide essential background to the Article but are not included in the printed version of the paper due to space constraints or being of interest only to a few specialists. A maximum of ten Extended Data display items (figures and tables) is typically permitted. When re-submitting your manuscript, please ensure that any supplementary figures and tables that are more critical to the manuscript's conclusions are converted to Extended data to increase these data's visibility.

Link Redacted

Note: This URL links to your confidential home page and associated information about manuscripts you may have submitted, or that you are reviewing for us. If you wish to forward this email to co-authors, please delete the link to your

homepage.

We hope to receive a suitably revised manuscript within 6 months. If you cannot send it within this time, please let us know. We will be happy to consider your revision so long as nothing similar has been accepted for publication at Nature Immunology or published elsewhere.

Nature Immunology is committed to improving transparency in authorship. As part of our efforts in this direction, we are now requesting that all authors identified as 'corresponding author' on published papers create and link their Open Researcher and Contributor Identifier (ORCID) with their account on the Manuscript Tracking System (MTS), prior to acceptance. ORCID helps the scientific community achieve unambiguous attribution of all scholarly contributions. You can create and link your ORCID from the home page of the MTS by clicking on 'Modify my Springer Nature account'. For more information please visit www.springernature.com/orcid.

Thank you for the opportunity to review your work.

Sincerely,

Ioana Staicu, Ph.D.
Senior Editor
Nature Immunology

Tel: 212-726-9207
Fax: 212-696-9752
www.nature.com/ni

Reviewers' Comments:

Reviewer #1 (Remarks to the Author):

In this manuscript, Frischholz et al. perform phenotypic and metabolic profiling of human antigen-specific CD8⁺ T cells responding to yellow fever virus (YFV) vaccination. They found that these antigen-specific CD8⁺ T cells are a heterogeneous population with discrete metabolic characteristics at different phases post vaccination. Among the different CD8⁺ T cells at the acute phase of the response, they found that central memory CD8⁺ T cells were the most proliferative population, corresponding to their increased metabolic dependence on glycolysis and especially oxidative phosphorylation (OXPHOS) to facilitate protein synthesis. Conversely, less activated naïve-like (TN(-like)) antigen-specific cells were quiescent based on their low proliferative and metabolic status and show a relative dependency on OXPHOS throughout their lifespan (i.e., at both the acute and memory phases post vaccination). Furthermore, at the memory phase, these TN(-like) antigen-specific CD8⁺ T cells represented the predominant population, suggesting that metabolic quiescence corresponds to improved persistence of these TN(-like) cells. These results help to unravel the temporal and subset-dependent metabolic processes (namely, glycolysis versus OXPHOS) that are associated with human memory T cell responses in a very robust vaccine setting. These findings may help uncover strategies to use modulate metabolism to boost memory T cell responses in less efficacious vaccine settings, which has important clinical implications. However, several concerns should be addressed prior to publication.

Major comments

1. The authors present compelling data for temporal and subset-specific metabolic dependencies in human antigen-specific CD8⁺ T cells responding to YFV vaccination. To address how broadly applicable such metabolic dependencies are for human T cell responses, the authors should perform one or more of the following:
 - a. The results should be validated in an independent antigen-specific system (e.g., in CD8⁺ T cells responding to other vaccinations such as influenza or SARS-CoV2).
 - b. The metabolic dependencies of other subsets of human antigen-specific CD8⁺ T cells could be profiled (e.g., those shown in Fig. S3C).
 - c. More detailed temporal analyses of metabolic dependencies of the different CD8⁺ T cell subsets after in vitro stimulation for various times (beyond the initial response at 24 hours) could be tested. This is important, because the vaccine-derived cells profiled in this study all likely have undergone some degree of proliferation (supported by the "proliferation index" analysis shown in Fig. 3A), whereas those stimulated with anti-CD3/anti-CD28 for 24 hours in vitro are expected to have undergone no or limited proliferation; therefore, the differences in metabolic dependencies may simply be due to differences in activation/differentiation states, which could confound the conclusions.
2. Related to the above point, mouse studies were performed at a single acute timepoint (day 6) and at the early memory phase (day 30/35). The inclusion of uninfected mice and additional timepoints at the early effector (e.g., days 2–4) and later memory timepoints (e.g., >day 60) after *Listeria monocytogenes* infection would be meaningful, since the human analyses

were performed across a broader spectrum of timepoints and activation states. It would also be beneficial to establish whether mouse antigen-specific CD8+ T cells from at least one additional infection model show disparate metabolic dependencies than human CD8+ T cells.

3. The authors show that the TN(-like) CD8+ T cells represent the predominant population at the memory phase of the vaccine response. Are TN(-like) cells more functional (based on expression of pro-inflammatory cytokines or cytotoxic molecules) upon antigen recall than the other populations, and if so, is this functionality impaired upon inhibition of OXPHOS (and not glycolysis)? Cells from acute and memory timepoints (versus unvaccinated) should be tested.

4. The authors seem to favor a model by which TN(-like) cells persist due to increased cell survival (based on the analysis of cleaved caspase-3). Do these cells also have the capacity to undergo self-renewal (e.g., homeostatic proliferation in response to IL-7 or IL-15 versus IL-2)? Is either the increased survival and/or possible self-renewal dependent on OXPHOS (and not glycolysis)?

5. Pseudotime analysis suggests that TN(-like) cells can progressively differentiate into the other subsets (i.e., central memory (Tcm), effector memory (Tem) and effector (TE) cells), terminating in the TE population (i.e., cluster 0) (Fig. S2F and G). Alternatively, it is possible that TN(-like) cells can generate all progeny of cells without undergoing such a linear, stepwise differentiation program. The authors should formally test such differentiation and the dependence on OXPHOS (and not glycolysis) and/or tone down their conclusions (e.g., by revising the models).

6. Despite performing single-cell transcriptomics profiling via CITE-seq, this dataset was minimally used to compare the metabolic features of bulk cycling cells and bulk TN(-like) cells (pooled from all timepoints). Several additional analyses would be meaningful to utilize the full power of this dataset and more comprehensively define metabolic differences between the various CD8+ T cell subsets/states responding to YFV vaccination:

- a. It would be meaningful to understand the metabolic pathways that are increased or decreased in TE, Tcm, and Tem populations versus the other populations (similar to the analyses performed in Fig. 3A).
- b. Given the dynamic regulation of metabolism that was revealed by protein translation rates at different timepoints after vaccination (Fig. 2D), it would be meaningful to compare the activities of metabolic signatures of the different T cell subsets at different timepoints, especially day 0, day 14 and 1 year (e.g., via violin plots).
- c. Based on the relative expression of marker genes (Fig. S2C) and proliferation index (Fig. 3A), TN(-like) cluster 2 appears to be more quiescent than TN(-like) cluster 1. Similarly, there are three unique clusters of Tcm with variable expression of quiescence-like and effector-like genes. Are there metabolic differences between these different cell states (based on gene signatures) at the peak effector phase? These analyses may help resolve why there is a large per-cell variability in metabolic state (see Fig. 2D) or metabolic dependencies (see Fig. 3C).
- d. Bioinformatics approaches to explore single-cell metabolic profiles, such as Vision (DeTomaso et al. 2019 Nat Commun; Ringel et al. 2020 Cell) or COMPASS analysis (Wagner et al. 2021 Cell), could also help parse out broader metabolic alterations (beyond OXPHOS and glycolysis) in the subsets.

Minor comments

1. The study seems to minimize previous work that has shown functional roles for both OXPHOS and glycolysis in memory T cells. For example, studies in mice and humans have shown that Tem have reduced OXPHOS/mitochondrial metabolic profiles compared to Tcm (Phan et al. 2016 Immunity; Nicoli et al. 2018 Front Immunol), and studies in mouse models have shown that increasing glycolysis (via VHL deletion) favors effector/memory and tissue-resident memory formation at the expense of central memory (Tcm) generation (Phan et al. 2016 Immunity; Liikanen et al. 2021 JEM). Further, despite their lower dependence on glycolysis at the peak effector timepoint than Tcm (as revealed in this study), human effector/memory CD8+ T cells require an early induction of glycolysis to support their effector functions, namely IFN-gamma production (Gubser et al. 2013 Nat Immunol), with studies in mouse models also supporting a functional role for glycolysis in shaping memory T cell function (Chang et al. 2013 Cell). Finally, studies in mouse models suggest that other glucose metabolism-related pathways (e.g., glyconeogenesis; Ma et al. 2018 Nat Cell Biol; Zhang et al. 2022 Mol Cell) are critical to support their formation and function. These points should be discussed.

2. In their CITE-seq transcriptomic profiling, the Leiden clusters do not reveal populations of Tscm cells that are readily distinguished based on the protein expression of markers, leading to differing proportions of the cells among the clusters based on transcriptomic and protein expression (compare Fig. 1G with S4B). To complement expression of individual genes, the authors could consider using TN, Tcm, Tem, and Tscm-like gene signatures to determine whether Tscm are distinguishable in the scRNA-seq dataset. Related to this, FAS expression should be depicted on the dot plot in Fig. S2C.

3. In Figure 1 legend key, please confirm that Table S8 is the correct callout. It should possibly be Table S1.

4. More description of the mathematical modeling shown in Figs. S7 and S8 would benefit the manuscript. These analyses and conclusions were challenging for a non-expert to understand.

Reviewer #2 (Remarks to the Author):

Schober and colleagues perform single cell studies to define the metabolic regulation of antigen-specific T cells in humans. By performing analyses on an interesting cohort of individuals vaccinated with the highly protective yellow fever vaccine, the

authors longitudinally follow antigen-specific CD8+ T cells by using single cell RNA-seq and the multidimensional flow cytometry-based SCENITH protocol. They identify a differential metabolic regulation of CD8+ T cell subsets. Despite the high technological tour de force applied to precious human samples, the main message of the manuscript remains a bit unclear to the reviewer. While description of dynamics and changes following perturbations (in this case, vaccination) are important in human studies, the highly correlative nature of the results prevents the identification of important implications for long-lived T cell memory.

Major comments

1. There is a major concern on the use of some assays on cryopreserved cells. Although very long studies on human fresh samples are not doable for obvious reasons, the paper reports in multiple instances inconsistencies between cryopreserved and fresh samples (puromycin assay, caspase 3 assay). Moreover, it is general knowledge that CD62L expression is generally lost on cryopreserved samples (PMID: 12957403; PMID: 15286731). It might be partially recovered following overnight rest in medium. However, distribution among specific T cell subsets will be affected due to differential apoptosis of these cells. Therefore, definition of memory subsets on the basis of CD62L on cryopreserved cells is a possible matter of concern.
2. As I perceive it, the main message the authors want to convey is that central memory T cells (TCM) are the most active phenotypic subset during response to yellow fever vaccine, while effector cells undergo metabolic shutdown. The implication of these differences in relation to long-lived T cell memory are unclear. In which way high basal protein synthesis should impact proliferation of TCM (page 5)? Could also the opposite be true? Results appear highly correlative and the implications of the findings are unclear in the absence of specific mechanistic investigations. These could be done on polyclonal memory T cells due to the paucity of antigen-specific subsets.
3. In some parts, results do not appear completely novel over previous literature. Figure 1 nicely confirms the dynamics of antigen-specific immune responses in response to vaccination, with some minor differences (e.g., definition of subsets, but please also see point 1; PMID: 18468462; PMID: 25855494). In Fig. 4, the authors report that tendency to apoptosis (cleaved caspase 3 in viability dye-negative cells) increases with progressive differentiation of antigen-specific T cells. This is somewhat expected and known from previous studies (PMID: 14625547; PMID: 23281401).
4. Fig. 4C relies on very few events per single subset, therefore quantification of cells in puromycin/caspase 3 gates is highly subjected to measurement errors. E.g., there is about 10 TCM cells at 1 year, thus every cell contributes about 10% to the quantification, thus certainly skewing results and statistics in Fig. 4D
5. Metabolic analysis would benefit from integration/correlation with scRNA-seq data and transcriptional activity/identity. The authors have a very precious dataset that would require more in depth analyses.

Minor concerns

1. At page 7, second paragraph, the authors state that "these ex vivo results differ from in vitro conditions in which stimulated T cells are more dependent on glycolysis and less in mitochondrial ATP production". It should be noted that activated T cells in vitro show a metabolic switch from OXPHOS to glycolysis, nevertheless they keep relying on OXPHOS at high rates (e.g., PMID: 22206904 Fig. 6 or PMID: 31747582 Fig. 1)

Version 1:

Decision Letter:

29th Oct 2025

Dear Drs. Schober and Frischholz,

Your Article, "Metabolic quiescence precedes and maintains human antigen-specific T-cell memory in vivo" has now been seen by 2 referees. Although we are interested in the possibility of publishing your study in Nature Immunology, the issues raised by the referees need to be addressed.

Please revise to address the remaining points from the referees. Please note that we agree with referee #3 regarding the fact that the Tn-like terminology can be confusing. Please define these cells very clearly at first description/first use and please use a designation that conveys the idea that these are a subset of memory T cells. We advise to use designations that follow the style of current ones for similar subsets (such as Tem, Tscm and so on). Could define them as naive-like memory T cells (Tnm cells).

At resubmission, please include a "Response to referees" detailing, point-by-point, how you addressed each referee comment (please specify the page number and the figures where the new data is found). If no action was taken to address a point, you must provide a compelling argument. This response will be sent back to the referees along with the revised manuscript.

Please include a revised version of any required reporting checklist. It will be available to referees to aid in their evaluation. <https://www.nature.com/documents/nr-reporting-summary.pdf>

When submitting the revised version of your manuscript, please pay close attention to our <https://www.nature.com/nature-portfolio/editorial-policies/image-integrity> Digital Image Integrity Guidelines and to the following points below:

-- that unprocessed scans are clearly labelled and match the gels and western blots presented in figures.

-- that control panels for gels and western blots are appropriately described as loading on sample processing controls
-- all images in the paper are checked for duplication of panels and for splicing of gel lanes.

Please note, Extended Data figures and tables are online-only (appearing in the online PDF and full-text HTML version of the paper), peer-reviewed display items that provide essential background to the Article but are not included in the printed version of the paper due to space constraints or being of interest only to a few specialists. A maximum of ten Extended Data display items (figures and tables) is typically permitted. When re-submitting your manuscript, please ensure that any supplementary figures and tables that are more critical to the manuscript's conclusions are converted to Extended data to increase these data's visibility.

Link Redacted

We hope to receive your revised manuscript within two weeks. If you cannot send it within this time, please let us know. We will be happy to consider your revision so long as nothing similar has been accepted for publication at Nature Immunology or published elsewhere.

Nature Immunology is committed to improving transparency in authorship. As part of our efforts in this direction, we are now requesting that all authors identified as 'corresponding author' on published papers create and link their Open Researcher and Contributor Identifier (ORCID) with their account on the Manuscript Tracking System (MTS), prior to acceptance. ORCID helps the scientific community achieve unambiguous attribution of all scholarly contributions. You can create and link your ORCID from the home page of the MTS by clicking on 'Modify my Springer Nature account'. For more information please visit www.springernature.com/orcid.

Sincerely,

Ioana Staicu, Ph.D.
Senior Editor
Nature Immunology

Tel: 212-726-9207
Fax: 212-696-9752
www.nature.com/ni

Reviewers' Comments:

Reviewer #1 (Remarks to the Author):

In this manuscript, Frischholz et al. have performed multiple new experiments and analyses of human SARS-CoV2 vaccination and mouse LCMV infection models, which have strengthened the overall conclusions of the study. Importantly, they also performed new experiments and identified basal protein synthesis and OXPHOS as key regulators of the stem-like state of naïve T cell (TN)-like cells in vitro, providing new mechanistic insight for how metabolism regulates the quiescence and self-renewal of this unique cell population that correlates with long-term protective immunity after yellow fever vaccination. They also performed additional analyses on their CITE-seq dataset. Overall, the authors have addressed my previous concerns, and I support its publication.

Reviewer #3 (Remarks to the Author):

Assessment of author rebuttal:

Reviewer 1

1a. The authors sufficiently responded to including an additional human antigen-specific system by adding past studies on

SARS-CoV-2 antigen-specific T cells (Figure S15 and S16).

1b. What the reviewer requested was not technically feasible since memory T cells at later time points are all metabolically quiescent.

1c. The authors included additional timepoints for in vitro analysis of metabolic phenotypes (Figure S11).

2. The authors added an additional mouse infection model (LCMV Armstrong) and assessed virus specific responses using P14 TCR transgenic T cells at early (day 6 and day 10) and later (day 30) time points (Figure S17). These experiments are important, since they demonstrated key differences between mouse and human T cell responses following in vivo challenge.

3. In response to the question of whether T_n(-like) cells are more functional after antigen recall, the authors indicated that this was already addressed in a previous publication (Fuentes Marraco, 2015). However, the authors proceeded to examine the responses of polyclonal T cells following in vitro stimulation of PBMCs with anti-CD3/anti-CD28/IL-2 using multiple time points (24, 48, 72 hours) and assessing CD8⁺ T cell subsets before and after stimulation (Figure S11), following short-term metabolic perturbation of anti-CD3/anti-CD28/IL-2 stimulated PBMCs (Figure S12), and following CD3/anti-CD28/IL-2 stimulation of sorted CD8⁺ T cell "memory" subsets (Figure S13). Although these experiments provided valuable insights into the in vitro differentiation dynamics and metabolic remodeling of CD8⁺ T cells subpopulations following polyclonal stimulation, the interpretation of these findings in the context of T_n(-like) cell biology was overstated. The populations classified as T_n(-like) in these studies were, in fact, composed predominantly of bona fide naïve T cells. Therefore, the observed T_n(-like) responses most likely represent authentic naïve T cell (T_n) responses. Can the authors comment on this?

4. The ability of T_n(-like) memory cells to undergo IL-7/IL-15 dependent homeostatic proliferation was not addressed by the authors but response to IL-15 stimulation was assessed in a previous publication.

5. The authors used a stepwise differentiation mathematical model, which provided a predictive framework that agreed with their experimental observations (Figure S7 and S8). However, they do acknowledge the limitations of this approach and that experimentally testing this model in humans is extremely challenging.

6a. The authors performed additional analysis on metabolic and immune pathways on their CITE-seq datasets. They added transcriptome heatmaps comparing: (i) time after vaccination, (ii) Leiden clusters from scRNA-seq datasets, (iii) CITE-seq. These re-analyses demonstrated that T_n(-like) cells possessed largely similar transcripts between scRNA and CITE-seq datasets (Figure 2b). The authors also further analyzed transcripts of antigen specific T cells in CM, EM, E subsets (Figure S9A). Their conclusions were not altered following reanalysis.

6b. Metabolic pathway activities were re-evaluated and visualized using row-normalized z-score plots. This analysis revealed that oxidative phosphorylation (OxPhos) signatures were consistently elevated across all examined time points (Figure 3 and S9). Furthermore, the authors demonstrated that quiescence-associated transcriptional programs were enriched in T_n(-like) and T_{cm} cell populations. This re-analysis further supported the authors original conclusion that quiescence and Oxphos are key signatures of T_n(-like) memory cells.

6c. The issue of discriminating T_n(-like) and T_{cm} subclusters were also addressed by the authors.

6d. The authors employed two additional bioinformatic approaches that were suggested by the reviewer to further characterize transcriptional metabolic profiles, namely Vision and COMPASS. As stated by the authors, Vision yielded similar results when compared to the bioinformatics tool they originally used, decoupleR.

Reviewer 2:

1. The reviewer expressed concerns on potential differences in experimental results between cryopreserved and fresh samples, in particular CD62L. The authors validated the appropriate use of the marker for identifying T_n(-like) cells by using scRNA-seq datasets from a previous publication (Figure S4D).

2. The authors clarified that the key finding in this manuscript is the metabolic characterization of T_n(-like) memory T cells. I think this is a major problem of the manuscript. The authors do not give enough background on what T_n(-like) memory cells are and their relevance in identified memory T cell subsets.

3. The reviewer raised concerns regarding the novelty of the findings presented in this manuscript. The authors responded by stating that combining metabolic readouts, such as SCENITH, with other modalities, further characterized a distinct CD8⁺ memory T cell subpopulation, T_n(-like) cells, that was described by the authors in a previous publication.

4. The authors made it clear that skewed interpretation of scRNA-seq data due to low event counts was mitigated by using aggregate data from multiple donors.

5. In depth analysis of scRNA-seq and Cite-seq datasets was already raised by reviewer 1 (point #6) and addressed by the authors.

A few specific things:

1. The authors mentioned a few times in the manuscript (e.g. Figure 4) and rebuttal, that they interrogated metabolic flux in antigen-specific T cells. However, SCENITH experiments do not examine 'metabolic flux' in cells, it assesses metabolic dependencies of protein translation. All mention of 'metabolic flux' in this context should be removed.
2. Although this may be considered a matter of semantics, I find the use of the term Tn(-like) to describe the quiescent memory T cell subset problematic. The inclusion of brackets around "-like" is unnecessary and potentially confusing. A more conventional nomenclature, such as Tn-like, would be clearer and less ambiguous. The current usage could be misinterpreted to suggest that the identified subset contains both naïve and memory T cells. This ambiguity is further compounded in Figures S11–S13, where the experiments appear to analyze bona fide naïve cells (CD62L+CD45RA+) rather than naïve-like memory cells, and yet the Tn(-like) term was still used.
3. Reviewer 2 expressed uncertainty regarding the principal finding of this manuscript. This confusion may, in part, stem from the current title. The metabolic characterization of Tn(-like) cells represents the central finding of the study and should be explicitly reflected in the title. Referring to "metabolic quiescence" without contextualizing it within the framework of Tn(-like) cell biology is potentially misleading, as bulk Tscm, Tcm, and even Tem populations also exhibit metabolic quiescence at later stages following antigenic challenge.

Version 2:

Decision Letter:

Our ref: NI-A40360B

14th Nov 2025

Dear Dr. Schober,

Thank you for submitting your revised manuscript "Metabolic quiescence of naïve-like memory T cells precedes and maintains antigen-specific T-cell memory" (NI-A40360B). We are happy to inform you that if you revise your manuscript appropriately according to our editorial requirements, your manuscript should be publishable in Nature Immunology.

I will now pre-edit the current version of your paper. We will also perform detailed checks on your paper and will send you a checklist detailing our editorial and formatting requirements in about two weeks. Please do not upload the final materials and make any revisions until you receive this additional information from us.

While waiting for the pre-edit check, please deposit all omic and code data into public repositories so that the accession codes are readily available to be added in the revised manuscript. We cannot accept the paper without the codes.

In addition, all corresponding authors need to update and link their ORCID to their Nature account. We cannot accept the paper without this information. We suggest that you look into this while waiting for the pre-edited manuscript. Should you have any query or comments about ORCID, please do not hesitate to contact our editorial assistant at immunology@us.nature.com.

If you had not uploaded a Word file for the current version of the manuscript, we will need one before beginning the editing process; please email that to immunology@us.nature.com at your earliest convenience.

Thank you again for your interest in Nature Immunology. Please do not hesitate to contact me if you have any questions.

Sincerely,

Ioana Staicu, Ph.D.
Senior Editor
Nature Immunology

Tel: 212-726-9207
Fax: 212-696-9752
www.nature.com/ni

Version 3:

Decision Letter:

In reply please quote: NI-A40360C

Dear Dr. Schober,

I am delighted to accept your manuscript entitled "Metabolic quiescence of naïve-like memory T cells precedes and maintains antigen-specific T-cell memory" for publication in an upcoming issue of Nature Immunology.

Over the next few weeks, your paper will be copyedited to ensure that it conforms to Nature Immunology style. Once your paper is typeset, you will receive an email with a link to choose the appropriate publishing options for your paper and our Author Services team will be in touch regarding any additional information that may be required.

Authors may need to take specific actions to achieve compliance with funder and institutional open access mandates. If your research is supported by a funder that requires immediate open access (e.g. according to [Plan S](https://www.springernature.com/gp/open-science/plan-s-compliance) principles or the [NIH public access policy](https://www.springernature.com/gp/open-science/us-federal-agency-compliance)) then you should select the gold OA route, and we will direct you to the compliant route where possible. Because authors warrant under our subscription licensing terms that they haven't committed to licensing any version of their article under a licence inconsistent with the terms of our agreement – including the applicable embargo period – publication under the subscription model isn't suitable for authors whose funders require no embargo.

Your paper will be published online soon after we receive your corrections and will appear in print in the next available issue.

Also, if you have any spectacular or outstanding figures or graphics associated with your manuscript - though not necessarily included with your submission - we'd be delighted to consider them as candidates for our cover. Simply send an electronic version (accompanied by a hard copy) to us with a possible cover caption enclosed.

If you have not already done so, we strongly recommend that you upload the step-by-step protocols used in this manuscript to [protocols.io](https://www.protocols.io). [protocols.io](https://www.protocols.io) is an open online resource that allows researchers to share their detailed experimental know-how. All uploaded protocols are made freely available and are assigned DOIs for ease of citation. Protocols can be linked to any publications in which they are used and will be linked to from your article. You can also establish a dedicated workspace to collect all your lab Protocols. By uploading your Protocols to [protocols.io](https://www.protocols.io), you are enabling researchers to more readily reproduce or adapt the methodology you use, as well as increasing the visibility of your protocols and papers. Upload your Protocols at <https://www.protocols.io>. Further information can be found at <https://www.protocols.io/help/publish-articles>.

Please note that we encourage the authors to self-archive their manuscript (the accepted version before copy editing) in their institutional repository, and in their funders' archives, six months after publication. Nature Portfolio recognizes the efforts of funding bodies to increase access of the research they fund, and strongly encourages authors to participate in such efforts. For information about our editorial policy, including license agreement and author copyright, please visit www.nature.com/ni/about/ed_policies/index.html

Sincerely,

Ioana Staicu, Ph.D.
Senior Editor
Nature Immunology

Tel: 212-726-9207
Fax: 212-696-9752
www.nature.com/ni

Click here if you would like to recommend Nature Immunology to your librarian
<http://www.nature.com/subscriptions/recommend.html#forms>

** Visit the Springer Nature Editorial and Publishing website at http://editorial-jobs.springernature.com?utm_source=ejP_NImm_email&utm_medium=ejP_NImm_email&utm_campaign=ejp_NImm for more information about our career opportunities. If you have any questions please click [here](mailto:editorial.publishing.jobs@springernature.com).

POINT-BY-POINT RESPONSE TO REVIEWS

Frischholz, Schuster, Grotz et al. (NI-A40360-T, "Metabolic quiescence precedes and maintains human antigen-specific T-cell memory in vivo")

(*Reviewer comments in italic, responses in non-italic*)

Overview of main manuscript changes during revision process

- Additional analyses of metabolic signatures in scRNAseq data of YFV vaccinees (new **Fig. 3; Fig. S9**)
- New metabolic analyses (both using SCENITH and scRNAseq) of human individuals after SARS-CoV-2 mRNA vaccination) (new **Fig. S15-16**)
- New investigation of a second mouse model – using P14 cells and LCMV – to complement results obtained in the *Lm*-OVA model with OT-I cells (extended **Fig. S17**)
- Extension of *in-vitro* SCENITH on polyclonal T cells with additional timepoints (**new Fig. S11**)
- New *in-vitro* analyses of proliferative capacity, differentiation and activation of phenotypic subsets of polyclonal T cells in conjunction with metabolic perturbation (**new Fig. S12-13**)
- Changes in the main text to accurately contextualize the novelty of the present work in light of the existing literature (introduction & discussion)

Reviewer #1

(*Remarks to the Author*)

In this manuscript, Frischholz et al. perform phenotypic and metabolic profiling of human antigen-specific CD8+ T cells responding to yellow fever virus (YFV) vaccination. They found that these antigen-specific CD8+ T cells are a heterogenous population with discrete metabolic characteristics at different phases post vaccination. Among the different CD8+ T cells at the acute phase of the response, they found that central memory CD8+ T cells were the most proliferative population, corresponding to their increased metabolic dependence on glycolysis and especially oxidative phosphorylation (OXPHOS) to facilitate protein synthesis. Conversely, less activated naïve-like (TN(-like)) antigen-specific cells were quiescent based on their low proliferative and metabolic status and show a relative dependency on OXPHOS throughout their lifespan (i.e., at both the acute and memory phases post vaccination). Furthermore, at the memory phase, these TN(-like) antigen-specific CD8+ T cells represented the predominant population, suggesting that metabolic quiescence corresponds to improved persistence of these TN(-like) cells. These results help to unravel the temporal and subset-dependent metabolic processes (namely, glycolysis versus OXPHOS) that are associated with human memory T cell responses in a very robust vaccine setting. These findings may help uncover strategies to use modulate metabolism to boost memory T cell responses in less efficacious vaccine settings, which has important clinical implications. However, several concerns should be addressed prior to publication.

We appreciate the reviewer's thoughtful and accurate summary of our key findings, as well as the recognition of the potential clinical relevance of our study. We are grateful for the constructive feedback and believe that we were able to address the raised points in a

meaningful manner. The revision has in our eyes strengthened both the clarity and the overall quality of the manuscript.

Major comments

1. *The authors present compelling data for temporal and subset-specific metabolic dependencies in human antigen-specific CD8⁺ T cells responding to YFV vaccination. To address how broadly applicable such metabolic dependencies are for human T cell responses, the authors should perform one or more of the following:*

a. The results should be validated in an independent antigen-specific system (e.g., in CD8⁺ T cells responding to other vaccinations such as influenza or SARS-CoV2).

We are glad about the reviewer's positive assessment of our dataset and simultaneously agree that extending our findings to additional human antigen-specific systems would enhance the impact and generalizability of our work. Our initial manuscript already provided a high-resolution analysis across numerous donors in a human system that often serves as a blueprint for successful immune responses (i.e., YFV vaccination). In addition, we could validate T-cell subset-dependent signatures of quiescence and activity in a murine *in-vivo* model (*Lm-OVA*) despite species-dependent differences in metabolic dependencies.

We do, however, recognize the value of further validation. For this reason, we first assessed metabolic signatures in antigen-specific CD8⁺ T cells using scRNAseq data from a SARS-CoV-2 mRNA vaccination cohort that we previously published (Kocher et al., 2024). Note that – for the revised version of the manuscript – we also re-analyzed metabolic signatures in the scRNAseq data from the YFV-vaccinated individuals, which we discuss in a response below. SARS-CoV-2 T-cell responses against three immunodominant spike epitopes (HLA-A*01:01/LTD, HLA-A*02:01/YLQ and HLA-A*03:01/KCY), which we detected with DNA-barcoded pHLA dextramers, all showed similar phenotypic kinetics (Kocher et al., 2024 and **Fig. S15A-B**).

Epitope-specific cells showed an elevated proliferation score at each acute timepoint after SARS-CoV-2 vaccination, at d10 after primary (“P1”), secondary (“S1”) and tertiary (“T1”) vaccination (**Fig. S15C**). At P1, epitope-specific T cells also showed elevated scores for OXPHOS and glycolysis, which were not as clearly boosted by subsequent vaccinations above the background level seen for unspecific cells or for epitope-specific cells at the memory timepoint (S3). We noted that the OXPHOS score is higher compared to the glycolysis score, similar as for YFV-specific cells. Whether or not the magnitude of the scores indicates the biological significance of a given pathway (i.e., whether a higher OXPHOS score proves that OXPHOS is more important compared to glycolysis) is not entirely clear, even though the scores were normalized to the number of genes contained in a score. We were surprised to see a boosting effect only for the proliferation score, but not for metabolic programs. Possibly, during recall responses, the transcriptional dynamics of the OXPHOS and glycolysis signatures are accelerated compared to the primary response, and day 10 after secondary vaccination therefore represents a – relatively speaking – “later” timepoint compared to day 10 after primary vaccination. This would be consistent with the concept of “translational preparedness” (Wolf et al., 2020). Moreover, previous studies (such as (Buck et al., 2016)) have demonstrated that memory T cells have a higher mitochondrial mass than naïve cells, suggesting that T-cell activation induces a metabolically primed state that facilitates T cells to metabolically react to restimulation without the need of a strong transcriptional response for these pathways.

Despite the differences in the vaccination settings and timepoints, we observed a high correlation of up- and downregulated pathways in epitope-specific T cells within the cycling cluster at d10 after secondary SARS-CoV-2 mRNA vaccination and d14 after a single YFV vaccination, with OXPHOS being among the top upregulated pathways (**Fig. S15D**).

Like in the YFV setting, we overall consider SCENITH analyses of metabolic flux to more accurately depict the metabolic profiles of epitope-specific T cells following immunization. We, therefore, next explored whether SCENITH-based metabolic flux analysis was possible in the same SARS-CoV-2 mRNA vaccination cohort. Despite limited cell numbers and scarce remaining samples from donors with clean antigen exposure histories (i.e., without breakthrough infections), we succeeded in performing SCENITH in T-cell populations specific for LTD and YLQ (**Fig. S16A-B**). Since our analysis on LTD-responses only comprised two donors (albeit with consistent phenotypes), we only report YLQ-responses. Please also note that sufficient samples and cell numbers for SCENITH were only available from d10 after second SARS-CoV-2 mRNA vaccination, while most of our YFV analyses focused on d14 after a single primary immunization.

Like in the YFV-vaccinated cohort, we used pHLA tetramers to identify epitope-specific T cells. Phenotypic subset distributions were similar compared to YFV-specific responses, with T_{EM} cells dominating (**Fig. S16C-D**). As seen with YFV, epitope-specific T_{CM} cells represented the most active subset (based on basal protein synthesis, BPS), while T_{EM} cells were less active than the average unspecific population, making also the total epitope-specific population appear less active (**Fig. S16E-F**). This confirms the importance of assessing metabolic flux on the subset level.

SARS-CoV-2-specific T cells showed a much higher dependence on mitochondrial respiration compared to glycolysis during the acute phase (**Fig. S16G**). For our analyses of T cells responding to SARS-CoV-2 mRNA vaccination, we could only use archived cryopreserved samples. With these samples, we again saw a strong association of T-cell differentiation degree and levels of cleaved Caspase 3 (**Fig. S16H-J**).

While these data were consistent with our previous results after YFV vaccination, the cell numbers of SARS-CoV-2 epitope-specific T_N -like cells (and in most cases also T_{SCM} and T_E cells) were too small to meet our quality cut-off for downstream analyses. Throughout the manuscript, we only performed downstream quantifications if at least 10 cells could be detected for a T-cell subset. This precluded a reliable assessment of whether SARS-CoV-2-specific T_N -like cells were also metabolically quiescent, although we expect them to be.

We had already in our published scRNAseq analyses of SARS-CoV-2-specific T cells (Kocher et al., 2024) observed much less T_N -like cells after vaccination compared to the YFV setting. This aligns with the general notion that the effective generation of quiescent T_N -like cells could be a distinguishing quality feature of the YFV vaccine, leading to decade-long T-cell memory after a single immunization (Akondy et al., 2017; Fuertes Marraco et al., 2015). Intriguingly, however, the metabolic features we did analyze (BPS, metabolic dependencies, cleaved Caspase 3) were highly reproducible for T_{CM} and T_{EM} cells in both vaccination cohorts, suggesting they represent universal biological features hardwired in distinct differentiation stages of T cells.

b. The metabolic dependencies of other subsets of human antigen-specific CD8+ T cells could be profiled (e.g., those shown in Fig. S3C).

The populations shown in **Fig. S3C** (with specificity to Influenza virus or SARS-CoV-2 epitopes) likely represent resting memory cells and would then be metabolically quiescent.

These populations are therefore not suitable for validating dynamic metabolic responses. Instead, our new analyses on SARS-CoV-2 specific T cells (see above) provide metabolic profiling of human antigen-specific CD8⁺ T cells in the context of another vaccination.

c. More detailed temporal analyses of metabolic dependencies of the different CD8⁺ T cell subsets after in-vitro stimulation for various times (beyond the initial response at 24 hours) could be tested. This is important, because the vaccine-derived cells profiled in this study all likely have undergone some degree of proliferation (supported by the “proliferation index” analysis shown in Fig. 3A), whereas those stimulated with anti-CD3/anti-CD28 for 24 hours in vitro are expected to have undergone no or limited proliferation; therefore, the differences in metabolic dependencies may simply be due to differences in activation/differentiation states, which could confound the conclusions.

We agree that extending our *in-vitro* stimulation assay beyond the 24-hour timepoint helps to distinguish activation state-dependent effects from those driven by proliferation. To this end, we performed new experiments and incorporated additional timepoints (48h and 72h next to 24h) to fully characterize the temporal evolution of metabolic dependencies *in vitro*, and to provide a clearer comparison with *in-vivo* vaccine responses. As in our initial analysis, polyclonal T cells were stimulated by plate-bound aCD3/aCD28 and low-dose IL-2 for different time intervals. At the end of each interval, SCENITH was performed (**Fig. S11A**).

Activation marker expression (CD69 and CD137) was highest 24h and 48h after activation and then slightly decreased (**Fig. S11B**). Ki-67 levels increased over time, and were highest in T_{CM} while T_N(-like) cells stayed negative for Ki-67 (**Fig. S11C-D**). BPS levels mirrored activation markers, being highest at 24h and decreasing almost to baseline at 72h (**Fig. S11E**). On a phenotypic subset level, the new replicates confirmed BPS to be highest in T_{SCM} at 24h, followed by T_{CM}. As *ex vivo* after YFV vaccination, T_{EM} and T_E cells after *in-vitro* stimulation contained puro^{lo} populations reflecting pre-apoptotic cells.

In terms of metabolic dependencies, glycolytic dependence decreased over time to levels which – at 72h – better reflected our *ex-vivo* observations after YFV vaccination (**Fig. S11F**). However, glycolytic dependence at 72h remained highest in T_{EM}, while at 24h it was highest in T_{CM}. Mitochondrial dependence was highest in least differentiated T_N(-like) cells, and overall did not markedly change over time (**Fig. S11G**).

Overall, these extended *in-vitro* SCENITH analyses confirm that T cells with an intermediate differentiation degree (here: T_{SCM} and T_{CM}) show highest metabolic activity, while T_N(-like) cells and T_E cells are more quiescent. In comparison to our *ex-vivo* findings, *in-vitro* stimulated cells were more dependent on glycolysis at 24h. Global metabolic differences at 72h did – in line with the reviewer’s suspicion – better reflect *ex-vivo* SCENITH, albeit *in-vitro* and *ex-vivo* results still differed on a subset level. We thank the reviewer for suggesting this informative revision experiment.

2. Related to the above point, mouse studies were performed at a single acute timepoint (day 6) and at the early memory phase (day 30/35). The inclusion of uninfected mice and additional timepoints at the early effector (e.g., days 2–4) and later memory timepoints (e.g., >day 60) after Listeria monocytogenes infection would be meaningful, since the human analyses were performed across a broader spectrum of timepoints and activation states. It would also be beneficial to establish whether mouse antigen-specific CD8⁺ T cells from at least one additional infection model show disparate metabolic dependencies than human CD8⁺ T cells.

We thank the reviewer for bringing up our analysis in the mouse model, which we believe provides valuable insights into both conserved and divergent aspects of CD8⁺ T-cell metabolism across species. In our initial experiments using *Listeria monocytogenes* (transgenic for OVA) infection (*Lm*-OVA), we focused on day 6 post-infection (d6 p.i.), as we considered this timepoint to best correspond to d14 in humans in terms of the acute effector response. In the revised version of the manuscript, we now report also additional timepoints in the acute effector phase (d8 and d10 p.i.), which demonstrate metabolic contraction kinetics compared to d6 (**Fig. S17**). While we recognize the interest in earlier timepoints (e.g., d2–4), practical limitations in recovering sufficient numbers of antigen-specific T cells precluded such analyses in our current system. Obtaining robust metabolic data at those early stages would require adoptive transfer of high numbers of TCR-transgenic T cells, which artificially elevates precursor frequencies and may distort physiological relevance.

For the revised version of the manuscript, we also examined metabolic profiles in a second murine infection model (**Fig. S17**). Specifically, we analyzed CD8⁺ T cells from Lymphocytic Choriomeningitis Virus (LCMV) Armstrong infected mice using P14 TCR-transgenic cells. This allowed us to evaluate whether the balance between activation-associated metabolic activity and quiescence is conserved across models, while also assessing whether the specific reliance on glycolysis or OXPHOS is species- or context-dependent.

The results obtained with the LCMV model were remarkably consistent with our analyses from the *Lm*-OVA model. In both models, at d6 p.i., TCR-transgenic T-cell populations were dominated by T_{EM} cells. At d8 and 10 p.i., T_E cells were equally dominant. Less differentiated KLRG1⁻ and more differentiated KLRG1⁺ T_{CM} cells were only present in small numbers at the acute phase of the response. However, KLRG1⁺ T_{CM} – together with T_{EM} cells – represented the metabolically most active subset at d6 p.i.; in contrast, KLRG1⁻ T_{CM} cells, were as quiescent as T_E cells. These results confirm our suspicion that KLRG1⁻ T_{CM} cells correspond to a more stem-like T_{CM} subset which is best comparable to T_N-like or T_{SCM} cells in the human system. Their quiescent state is also perfectly compatible with previous work by others (Bresser et al., 2022), which showed that KLRG1⁻ T_{CM} cells do not accumulate large proliferative histories while KLRG1⁺ T_{CM} cells proliferate most. Overall, the BPS results from both models also strongly align with our human data, with early/intermediate/late differentiated cells showing a healthy-quiescent/active/unhealthy-quiescent state, respectively.

Analysis of the d8 and d10 p.i. timepoints in both models revealed that metabolic activity subsided quickly and reached a similarly quiescent state across all subsets, at a level that did not change until later memory at d30 p.i.. At this later timepoint after LCMV infection, KLRG1⁻ P14 T_{CM} cells became more frequent, like OT-I cells after *Lm*-OVA infection. This again resembled the dynamics we see for human T_N-like cells.

Lastly, we evaluated metabolic dependencies of murine T cells on glycolysis and mitochondrial respiration. Again, the results were remarkably consistent between the models. Glycolytic dependence was much higher than for human T cells, at a level of around 50% both in the acute and the memory phase and without major differences between subsets. Mitochondrial dependence, conversely, was lower than we observed for human T cells, at a similar average level as seen for glycolysis both in the acute and memory phases. Interestingly, KLRG1⁺ T_{CM} and T_{EM} cells showed less mitochondrial dependence than other subsets at d6 p.i. (when they were the most active subsets based on BPS).

Overall, our previous analyses in the *Lm*-OVA model and the new results from the LCMV model are not only consistent in direct comparison, but also align with our human data in terms of the frequencies and activity states of phenotypic subsets. Metabolic dependencies, in contrast, showed species-specific features, demonstrating the importance of standardized investigation of basic T-cell biology in human model systems.

3. The authors show that the TN(-like) CD8⁺ T cells represent the predominant population at the memory phase of the vaccine response. Are TN(-like) cells more functional (based on expression of pro-inflammatory cytokines or cytotoxic molecules) upon antigen recall than the other populations, and if so, is this functionality impaired upon inhibition of OXPHOS (and not glycolysis)? Cells from acute and memory timepoints (versus unvaccinated) should be tested.

4. The authors seem to favor a model by which TN(-like) cells persist due to increased cell survival (based on the analysis of cleaved caspase-3). Do these cells also have the capacity to undergo self-renewal (e.g., homeostatic proliferation in response to IL-7 or IL-15 versus IL-2)? Is either the increased survival and/or possible self-renewal dependent on OXPHOS (and not glycolysis)?

(response to points 3 and 4)

We thank the reviewer for highlighting these important points. It has been previously demonstrated that naïve-like YFV NS4B-specific CD8⁺ T cells possess strong *in-vitro* expansion capacity (Fuentes Marraco et al., 2015). Consistently, our data suggest that quiescent, early-differentiated cells after YFV vaccination reconstitute long-term memory responses and, thus, are rather important for re-expansion than fast expression of effector molecules. In line with this hypothesis, it has been demonstrated that cytokines and granzymes are mostly expressed by intermediate differentiated antigen-experienced human CD8⁺ T cells after antigen-independent restimulation *in vitro*, while early-differentiated, antigen-experienced cells have the highest capacity to produce IL-2 and to expand (Zwijnenburg et al., 2023).

We followed the reviewer's advice and investigated this issue further. This enabled us to perform more mechanistic investigation of how proliferation, differentiation and activation depend on metabolic pathways as well as on protein translation. Due to the paucity of antigen-specific T cells following YFV vaccination (especially at late memory timepoints), these analyses were only possible with polyclonal primary human T cells.

For the revised version of the manuscript, we stained polyclonal primary human T cells from healthy donors with Cell Trace Far Red (CTFR), and stimulated them with aCD3/aCD28 and low-dose IL-2 (**Fig. S12A**). Simultaneously, cells were treated with harringtonine, 2-DG, oligomycin or no inhibitor (control condition) for the initial 24h of stimulation. After a final stimulation time of 24h, 48h or 72h, flow cytometric analysis was performed. In contrast to the extended *in-vitro* SCENITH experiment that we also performed (**Fig. S11**), this experiment thus investigated the effect of early yet longer (24h) inhibition of metabolism on later proliferation, differentiation and activation.

Upon stimulation, the relative share of T_{SCM} and T_{CM} cells increased over time, while the fraction of TN(-like) cells decreased (**Fig. S12B**). Blockade of protein translation, but also of OXPHOS or glycolysis in particular, inhibited T_{CM} outgrowth (**Fig. S12C**). Instead, T cells were either stalled at the TN(-like) stage or underwent accelerated differentiation into T_E cells in the presence or absence of stimulation.

Proliferation – as assessed by dilution of CTFR – started to become visible only at 48h, and more robustly occurred at 72h with most cells having undergone more than two cell divisions (**Fig. S12D**). T_{CM} cells displayed the highest degree of proliferation, while the majority of TN(-like) cells only underwent a maximum of one cell division within 72h (**Fig. S12E**). This was consistent with prior literature (Gattinoni et al., 2011), our own *in-vivo* results and Ki-67 staining of *in-vitro* activated T cells (**Fig. S11**), further demonstrating the quiescence of this stem-like subset.

Initial harringtonine treatment (and thus global inhibition of protein translation) completely abolished proliferation in all phenotypic subsets, which was almost to the same degree visible after inhibition of OXPHOS through oligomycin (**Fig. S12F**). In terms of absolute cell numbers, we observed an initial cell loss at 24h likely due to activation-induced cell death (**Fig. S12G**). At 72h – coincidental with proliferative activity taking off – cell numbers were higher than starting numbers. When normalized to control-treated cells, cell numbers were negatively affected by all treatments upon stimulation (**Fig. S12H**). However, intriguingly, initial 2-DG treatment did not affect cell counts in the absence of stimulation. Overall, these results mirrored our *in-vivo* findings that glycolysis has a particular role for the anabolism of proliferating cells.

While these findings proved insightful, subset analysis was restricted to the phenotypes that the cells had at a given time, and may be confounded by differentiation occurring *in vitro*. Therefore, we next performed two independent experiments for which we sorted polyclonal T_N(-like) (input population 1), T_{SCM}+T_{CM} (input population 2) or T_{EM}+T_E (input population 3) cells, and stimulated them with aCD3/aCD28 and low-dose IL-2 for 72h (**Fig. S13A**). Simultaneously, cells were treated with harringtonine, 2-DG, oligomycin or no inhibitor (control condition) for the initial 24h. After 72h of activation, flow cytometric analysis was performed.

Cell counts after inhibitor treatment mirrored the results from our previous experiments (**Fig. S13B**). We confirmed that in the absence of stimulation – for all subsets – early inhibition of glycolysis through 2-DG treatment had no effect on cell counts, while inhibition of OXPHOS and overall protein translation had a negative effect. In the stimulated condition, T_N(-like) sorted cell numbers were more dependent on OXPHOS, while T_{SCM}+T_{CM} and T_{EM}+T_E sorted cells seemed more dependent on glycolysis.

Initially, T_{SCM}+T_{CM} sorted cells mainly consisted of T_{CM} cells while T_{EM}+T_E sorted cells largely consisted of T_{EM} cells (**Fig. S13C**). In the control conditions, input populations differentiated into downstream progenitors, with T_N(-like) sorted cells demonstrating pluripotency and the highest relative share of maintained early differentiated populations. As described before (Geginat et al., 2003; Lugli et al., 2013), T_E cells were poorly induced or maintained under these *in-vitro* conditions.

T_N(-like) sorted cells that were initially treated with harringtonine showed more differentiation upon stimulation into T_{EM} or T_E cells, and less “stalling” at the T_N(-like) stage (**Fig. S13D**). This effect was even more pronounced upon initial oligomycin treatment, suggesting that early protein translation and OXPHOS in T_N(-like) cells are important for preservation of a stem-like state. This is consistent with the concept of “translational preparedness” of T_N(-like) cells (Wolf et al., 2020). While the link of OXPHOS and stemness is well established, the dependence on protein translation also suggested that quiescence is a state that needs to be actively maintained. In other words, when T_N(-like) sorted cells relied solely on glycolysis during initial activation (when OXPHOS is blocked through oligomycin), these cells underwent accelerated differentiation towards the T_E stage (**Fig. S13D**, also see **Fig. S12C**). This setting, which is similarly used in a Seahorse assay to investigate the “glycolytic reserve”, thereby revealed the anabolic function of glycolysis we and others suggest. Notably, T_{SCM}+T_{CM} or T_{EM}+T_E sorted cells – which were already more differentiated to begin with – did not show such prominent dependencies on initial metabolism in terms of their further differentiation into downstream subsets upon stimulation. In the absence of stimulation, however, maintenance of the T_{CM} state was also dependent both on OXPHOS and protein translation (**Fig. S13D**).

In terms of activation, T_N(-like) sorted cells showed less upregulation of CD69 and CD137 compared to T_{SCM}+T_{CM} or T_{EM}+T_E sorted cells following stimulation (**Fig. S13E**). This mild upregulation of activation markers may have been confounded by FAS-mediated inhibitory effects among T_N(-like) cells (Klebanoff et al., 2016). However, all subsets were completely dependent on initial protein translation to upregulate activation markers later at 72h.

Overall, these results underline the quiescent and stem-cell like characteristics of T_N(-like) cells. To answer the reviewer's questions from point 3 and 4 most directly, these features, rather than effector function (here shown by upregulation of activation markers) distinguish T_N(-like) cells from the other subsets. Our new *in-vitro* analyses confirm that T_N(-like) cells were able to do self-renewal next to differentiation into downstream effector subsets. Of note, our novel data also show that T_N(-like) cells strongly relied on OXPHOS and protein translation to actively maintain their cell numbers as well as their T_N(-like) phenotype – both in the presence and absence of stimulation. We think these analyses significantly improved our manuscript, and we are thankful for the reviewer's suggestion.

5. Pseudotime analysis suggests that TN(-like) cells can progressively differentiate into the other subsets (i.e., central memory (Tcm), effector memory (Tem) and effector (TE) cells), terminating in the TE population (i.e., cluster 0) (Fig. S2F and G). Alternatively, it is possible that TN(-like) cells can generate all progeny of cells without undergoing such a linear, stepwise differentiation program. The authors should formally test such differentiation and the dependence on OXPHOS (and not glycolysis) and/or tone down their conclusions (e.g., by revising the models).

We agree with the reviewer that definitively demonstrating a stepwise differentiation trajectory in human T cells remains highly challenging. While numerous murine studies support a progressive transition from naïve-like to more differentiated subsets (Buchholz et al., 2016), we recognize that alternative models have been proposed and that definitive lineage-tracing experiments are not feasible in human systems. In our study, we adopted the stepwise model to inform our mathematical modeling analysis, and it indeed provided a coherent and predictive framework that aligned well with our experimental observations. However, we acknowledge that this remains a model rather than direct proof of lineage progression. To address the reviewer's concern, we revised the manuscript text to more clearly communicate that a stepwise differentiation model remains difficult to prove in human studies even if it is consistent with our data and supported by prior literature. We thereby still explicitly acknowledge the potential for alternative differentiation trajectories:

(in reference to mathematical modeling results)

“Of note, these modeling results do not formally exclude the possibility that less differentiated subsets like T_N(-like) cells may directly differentiate into subsets like T_{EM} cells without a linear, stepwise differentiation program.”

6. Despite performing single-cell transcriptomics profiling via CITE-seq, this dataset was minimally used to compare the metabolic features of bulk cycling cells and bulk TN(-like) cells (pooled from all timepoints). Several additional analyses would be meaningful to utilize the full power of this dataset and more comprehensively define metabolic differences between the various CD8+ T cell subsets/states responding to YFV vaccination: a. It would be meaningful to understand the metabolic pathways that are increased or decreased in TE, Tcm, and Tem populations versus the other populations (similar to the analyses performed in Fig. 3A). b. Given the dynamic regulation of metabolism that was revealed by protein translation rates at different timepoints after vaccination (Fig. 2D), it would be meaningful to compare the activities of metabolic signatures of the different T cell subsets at different timepoints, especially day 0, day 14 and 1 year (e.g., via violin plots). c. Based on the relative expression of marker genes (Fig. S2C) and proliferation index (Fig. 3A), TN(-like) cluster 2 appears to be more quiescent than TN(-like) cluster 1. Similarly, there are three unique clusters of Tcm with variable expression of quiescence-like and effector-like genes. Are there metabolic differences

between these different cell states (based on gene signatures) at the peak effector phase? These analyses may help resolve why there is a large per-cell variability in metabolic state (see Fig. 2D) or metabolic dependencies (see Fig. 3C). d. Bioinformatics approaches to explore single-cell metabolic profiles, such as Vision (DeTomaso et al. 2019 Nat Commun; Ringel et al. 2020 Cell) or COMPASS analysis (Wagner et al. 2021 Cell), could also help parse out broader metabolic alterations (beyond OXPHOS and glycolysis) in the subsets.

We thank both reviewers for raising this point. We had indeed considered the potential of our scRNAseq dataset for metabolic profiling but initially decided to focus on SCENITH-derived measurements of metabolic flux, which we view as more functionally informative. RNA expression levels alone do not always reflect protein levels and overall pathway activity.

However, we do agree that incorporating transcriptomic analysis of metabolic signatures enhances the manuscript and provides a more balanced and comprehensive assessment. In the revised version, we therefore integrated in-depth analyses of our scRNAseq data to assess glycolysis and OXPHOS pathway activity, as well as broader metabolic and immunological programs (new **Fig. 3**). We thereby explored the use of advanced computational tools such as Vision and COMPASS, as recommended by the reviewer (sub-point d), which use weighted gene scores and thereby counteract some of the concerns phrased above by the reviewer. Using Vision (DeTomaso et al., 2019), we re-performed transcriptional pathway analyses that we had done before using decoupleR (Badia-I-Mompel et al., 2022) and also performed new analyses. Vision often yielded similar results as decoupleR, but in some instances more consistently matched our flow cytometry-based results.

Ad a. Following the reviewer's suggestion, we employed Vision to re-analyze our YFV-scRNAseq dataset with a specific focus on metabolic and immunological pathways. This renewed analysis did not strongly change our findings for T_N(-like) and cycling cells compared to our previous analysis using decoupleR (see **Fig. 3A**). Further, we now also included similar Volcano Plots for the other combined Leiden clusters of T_{CM}, T_{EM} and T_E cells according to the reviewer's suggestion (**Fig. S9A**), and present heatmaps visualizing selected pathway signatures across the most relevant Leiden clusters (**Fig. 3B**). For a detailed discussion see below the answer to point b.

Ad b. To obtain a comprehensive overview beyond our initial analyses (**Fig. 3A**), we investigated the mean scores for individual pathways (see rows in **Fig. 3B** and **Fig. S9C**), and plotted them for A2/NS4B-specific T cells as row-normalized z-scores per timepoint, per selected Leiden cluster or per CITEseq phenotype. This demonstrated that quiescence is downregulated in the acute phase of the immune response, in which OXPHOS and glycolysis are prominent. Absolute pathway scores (see horizontal bars on the right in **Fig. 3B** and **Fig. S9C**) are based on overall transcript levels per gene in a given pathway. Assuming that these scores reflect the biological significance of a pathway, OXPHOS signatures were relatively more prominent than glycolysis signatures even at acute timepoints. Generally, the pathway scores correlate both with absolute metabolic activity and relative dependence on the pathway, the two of which cannot be distinguished. Note that, in comparison, SCENITH measures absolute activity through BPS and relative dependencies on specific pathways separately.

In terms of phenotypic subsets, quiescence signatures were most prominent in clusters and CITEseq populations corresponding to T_N(-like) cells or non-cycling T_{CM} cells (**Fig. 3B**). In cycling cells, OXPHOS and glycolysis were both upregulated compared to other Leiden clusters, although the absolute pathway scores suggest a larger role for OXPHOS as explained above. Please note that cycling cells represented the majority of cells at d14 and mostly had a T_{CM} and T_{EM} CITEseq phenotype (**Fig. 1I, S4B**). Non-cycling T_{CM1} and T_{CM2} cells were relatively more active in glycolysis, while OXPHOS was transcriptionally more engaged by T_N(-like) cells (**Fig. 3B**). This was confirmed on the CITEseq level. For the core pathways

OXPHOS, glycolysis and T-cell proliferation, we also visualized longitudinal timelines, UMAPs and violin plots, as suggested by the reviewer (**Fig. 3C-E**). These data indicate that, time-wise, OXPHOS genes were increased in A2/NS4B⁺ cells compared to A2/NS4B⁻ cells in the cycling cluster at d14, but similarly low again at one year (**Fig. 3C**). Smaller (on an absolute, but not relative level) yet consistent differences were noticeable for the glycolysis score. Both metabolic scores showed an earlier peak (d11) than the proliferation score (d14), which mirrored our flow cytometric data with BPS peaking before activation markers or Ki-67 (**Fig. S5C, S6F, S6K**). On a cluster level, OXPHOS genes were highest in the cycling cluster, also high in naïve-like clusters and lowest in the more differentiated T_{EM} and T_E clusters (**Fig. 3D-E**). The glycolysis score showed similar patterns, but was also elevated in non-cycling T_{CM} but not in T_N(-like) cells.

Our new analyses yielded several additional findings. For example, pathways for apoptosis and base excision repair followed similar longitudinal kinetics and were both expectedly upregulated in cycling cells, but showed key differences in the distribution across transcriptional clusters and CITEseq subsets (**Fig. 3B**). Apoptosis was upregulated in T_{EM} and T_E clusters with IFN signatures, indicating contraction in these populations during the “cooling phase” of the immune response. In contrast, base excision repair was upregulated in the T_N(-like) clusters, indicating that stem-like cells are transcriptionally prepared for genomic surveillance without showing elevated levels of γH2Ax protein in the absence of DNA damage (compare to **Fig. 5F**). Finally, we also explored additional metabolic pathways for patterns with regards to longitudinal dynamics or subset distribution. This led to novel observations, for example an upregulation of linoleic acid metabolism (Nava Lauson et al., 2023) specifically in the memory phase, while arachidonic acid signatures (Bibby et al., 2022) were prominent in the early acute (d7) as well as in the memory phase. We report these data as an outlook and resource for future investigation (**Fig. S9C**).

Ad c. We thank the reviewer for raising the point of different T_N(-like) or T_{CM} clusters present in our analysis. Upon careful re-analysis, we noticed that cluster 11 (formerly labeled as T_{CM}3) actually corresponds to KIR⁺ CD8⁺ T cells (Li et al., 2022). We re-annotated this cluster as such and excluded it from downstream analyses of grouped T_{CM} Leiden clusters. The KIR⁺ cluster contained, across all timepoints and donors, a total of only 7 NS4B-specific cells. Our error therefore did not alter any of our findings, but we thank the reviewer for bringing this to our attention and apologize for the confusion.

The other T_N(-like) or T_{CM} clusters did differ in terms of phenotypes and we already reported this in the original manuscript on several occasions. For example, cluster T_N(-like)1 was indeed – as the reviewer suspected – slightly higher in glycolysis or T-cell activation and proliferation scores, whereas it also showed a higher quiescence score (**Fig. 3B**). Compared to T_{CM}2 cells, T_{CM}1 cells were higher in quiescence and glycolysis, but lower in T-cell activation and proliferation. However, we cannot pinpoint any decisive factors that would justify classifying these clusters as unique subsets on a meta-level. In fact, when directly comparing the pathways that are enriched or depleted in those clusters versus all other clusters, there was a strong correlation between the two naïve and the two T_{CM} clusters (**Fig. R1**).

Fig. R1: Comparison of metabolic pathway transcription in naïve and central memory CD8⁺ T-cell clusters. Differential pathway analysis of A2/NS4B⁺ T cells at d14 post YFV vaccination, comparing (A) naïve clusters T_{N(-like)1} and T_{N(-like)2} or (B) central memory clusters T_{CM1} and T_{CM2}. Left: volcano plot of metabolic and cellular pathway scores in (A) T_{N(-like)1} vs. T_{N(-like)2} or (B) T_{CM1} vs. T_{CM2}. Black: significantly changed pathways (p<0.05). Grey: non-significant changes. Right: pathway correlation between clusters (A) T_{N(-like)1} and T_{N(-like)2} or (B) T_{CM1} and T_{CM2} at d14 filtered for pathways significantly enriched or depleted (p<0.05) in the indicated cluster over all other clusters.

In our eyes, the more important distinction is to separate these non-cycling T_{CM1} and T_{CM2} cluster cells from the T_{CM} cells present in the cycling cluster. Our flow cytometry data indicated that T_{CM} cells consisted of a proliferative and metabolically active subpopulation in parallel to resting T_{CM} cells (see, e.g., **Fig. 2C** or **Fig. 5C**). In the scRNAseq data, we found that, at d14, the majority of CITEseq-defined T_{CM} cells were located within the cycling cluster (**Fig. S9D**). Comparison of cycling and non-cycling CITEseq-defined T_{CM} cells showed that pathways associated with T-cell activation and proliferation, but also OXPHOS, were specifically enriched in cycling T_{CM}, while ribosomal genes and the quiescence score were enriched in non-cycling T_{CM} cells (**Fig. S9E**). Next to T_{CM}, the cycling cluster also comprised CITEseq-defined T_{EM} cells (**Fig. S4B**). Importantly, comparing the enriched/depleted pathways in T_{CM} and T_{EM} cells within the cycling cluster revealed that cycling T_{CM} cells were more active and proliferative with a more pronounced transcriptional OXPHOS signature than cycling T_{EM} cells (**Fig. S9F**). This recapitulated our flow cytometry-derived metabolic data.

In summary, phenotypic subsets displayed distinct transcriptional metabolic programs during the course of a human T-cell response. While quiescence was associated with stem-like T cells and thus enriched at d0 and at late memory timepoints, key metabolic pathways peaked at d11 in cycling cells and suggested a high relevance for OXPHOS during this acute phase.

Minor comments

1. The study seems to minimize previous work that has shown functional roles for both OXPHOS and glycolysis in memory T cells. For example, studies in mice and humans have shown that Tem have reduced OXPHOS/mitochondrial metabolic profiles compared to Tcm (Phan et al. 2016 Immunity; Nicoli et al. 2018 Front Immunol), and studies in mouse models have shown that increasing glycolysis (via VHL deletion) favors effector/memory and tissue-resident memory formation at the expense of central memory (Tcm) generation (Phan et al.

2016 *Immunity*; Liikanen et al. 2021 *JEM*). Further, despite their lower dependence on glycolysis at the peak effector timepoint than T_{CM} (as revealed in this study), human effector/memory CD8⁺ T cells require an early induction of glycolysis to support their effector functions, namely IFN- γ production (Gubser et al. 2013 *Nat Immunol*), with studies in mouse models also supporting a functional role for glycolysis in shaping memory T cell function (Chang et al. 2013 *Cell*). Finally, studies in mouse models suggest that other glucose metabolism-related pathways (e.g., glyconeogenesis; Ma et al. 2018 *Nat Cell Biol*; Zhang et al. 2022 *Mol Cell*) are critical to support their formation and function. These points should be discussed.

We thank the reviewer for pointing this out and want to clarify that it was never our intention to minimize the importance of prior studies demonstrating key roles for both OXPHOS and glycolysis in memory and effector T-cell function. We cited several foundational studies (see, e.g., references 16–24 in the revised manuscript) in our introduction. These and other studies that we had already cited initially included references mentioned by the reviewer (Chang et al., 2013; Nicoli et al., 2018; Phan et al., 2016). In the revised version of the manuscript, we now also cited additional studies that the reviewer pointed out (Gubser et al., 2013; Liikanen et al., 2021; Ma et al., 2018; Zhang et al., 2022). This entire prior work has greatly informed our understanding of T-cell metabolism. That said, to the best of our knowledge, direct *ex-vivo* measurements of metabolic flux at the level of defined antigen-specific CD8⁺ T-cell subsets had not previously been performed in either mice or humans. Our study aimed to complement and build upon earlier analyses by providing this missing dimension. In the revised manuscript, we more thoroughly referenced and discussed the important studies cited by the reviewer, and more clearly articulated how our findings extend and integrate with this existing body of work. We also tried to more clearly highlight the unique contribution of our datasets in allowing *ex-vivo* subset-resolved profiling of metabolic activity using SCENITH and single-cell sequencing.

Example from introduction: “It has been known for more than five decades that T cells become metabolically active after *in-vitro* activation, with elevated rates of glycolysis, and that resting cells are quiescent, feeding on mitochondrial respiration (Chang et al., 2013; Pearce et al., 2009; Roos and Loos, 1973; van der Windt et al., 2012). Yet, recent mouse studies made clear that *in-vivo* activated T cells maintain a high dependence on OXPHOS even during rapid expansion in the acute phase of an immune response (Frisch et al., 2025; Ma et al., 2019). In mice, genetic knockouts allow mechanistic investigation more easily compared to studies in humans (Liikanen et al., 2021; Phan et al., 2016; Sukumar et al., 2013), but have led to conflicting observations on the interplay of metabolism and T-cell memory formation. T cell-specific knockout of Von-Hippel Lindau stabilizes HIF-1 α , elevates glycolysis and reduces OXPHOS. This leads to accelerated differentiation of T_{CM} cells into more differentiated T_{EM} cells, yet retains functional memory formation despite a decreased respiratory metabolism (Liikanen et al., 2021; Phan et al., 2016). Conversely, overexpression of the glycolytic enzyme phosphoglycerate mutase 1 impairs T-cell memory formation, while memory is enhanced by inhibition of glycolysis through 2-deoxy-D-glucose (2-DG) (Sukumar et al., 2013). Despite this important prior work, overall, the precise metabolic states of memory precursors *in vivo* require better definition. Population-level assessments fail to capture subset-specific metabolic dependencies, underscoring the need for direct measurements of antigen-specific T-cell metabolism at subset- or single-cell resolution throughout the immune response. This particularly applies to studies in humans.”

Example from discussion: “The long-standing dogma that activated T cells are highly glycolytic and resting cells depend more on OXPHOS has been challenged in recent years. For example, murine T cells during acute responses *in vivo* critically engage OXPHOS and use glycolysis (as well as glycogenolysis) in the acute but also memory phases for anabolic needs and to control reactive oxygen species (Ma et al., 2018; van der Windt and Pearce, 2012; Zhang et

al., 2022), highlighting major differences to *in-vitro* results (Levine et al., 2021; Ma et al., 2019). Our study demonstrates the importance of OXPHOS for energy production throughout long-lasting human immune responses *in vivo*. Importantly, our data reveal this strong metabolic dependence on OXPHOS to prevail among all phenotypic subsets and across acute and memory timepoints. Glycolysis, in contrast, was only upregulated in the acute phase, to a low extent, and skewed to the most proliferative T_{CM} subset. We thereby uncover for human antigen-specific T cells primed *in vivo* that energy generation (catabolism) is most dependent on OXPHOS, whereas the association of activity and mild glycolytic dependence in T_{CM} cells suggests a role for anabolism in this T-cell subset. This aligns with studies on polyclonal human T_{EM} cells or murine T cells relying on glycolysis to switch on cytokine-producing effector function (Chang et al., 2013; Gubser et al., 2013). It also fits to observations of T_{EM} cells possessing reduced mitochondrial metabolic profiles compared to T_{CM} cells (Nicoli et al., 2018; Phan et al., 2016).”

2. In their CITE-seq transcriptomic profiling, the Leiden clusters do not reveal populations of Tscm cells that are readily distinguished based on the protein expression of markers, leading to differing proportions of the cells among the clusters based on transcriptomic and protein expression (compare Fig. 1G with S4B). To complement expression of individual genes, the authors could consider using TN, Tcm, Tem, and Tscm-like gene signatures to determine whether Tscm are distinguishable in the scRNA-seq dataset. Related to this, FAS expression should be depicted on the dot plot in Fig. S2C.

We agree with the reviewer that T_{SCM} cells did not form a clearly defined transcriptional cluster in our dataset, in contrast to other subsets more easily resolved by protein markers. We find this discrepancy between protein-based and transcriptomic clustering to be a biologically interesting observation. According to the reviewer’s suggestion, we have generated gene signatures for T_N(-like), T_{SCM}, T_{CM}, and T_{EM} cell states from published data (Gattinoni et al., 2011) to further explore whether T_{SCM} populations can be distinguished at the transcriptomic level (new **Fig. S2F**). These analyses validated our previous categorization of cell types, and confirmed that T_{SCM} cells, in contrast to the other subsets, cannot be as stringently assigned to distinct transcriptional states (i.e., clusters).

Regarding *FAS* expression, we did show this in **Fig. S4A** alongside CD95 protein levels. However, we gladly added *FAS* expression to the dot plot in **Fig. S2C** to highlight this marker more clearly. *FAS* expression was highest in the “Ribo⁰” clusters, which likely represent pre-apoptotic cells. Among the other subsets, *FAS* expression, like the T_{SCM} signature, was more broadly distributed. Next to *FAS*, we also added *HNRNPLL*, which encodes the enzyme that splices CD45 isoforms (**Fig. S2C**).

3. In Figure 1 legend key, please confirm that Table S8 is the correct callout. It should possibly be Table S1.

We thank the reviewer for pointing this out and apologize for the oversight. The correct reference in the **Fig. 1 legend** should have been **Table S9**, not **Table S8**. We changed this now to **Table S2** in the revised version to follow the chronological order in which they are referenced in the manuscript and also updated the numbering of all other supplementary tables.

4. More description of the mathematical modeling shown in Figs. S7 and S8 would benefit the manuscript. These analyses and conclusions were challenging for a non-expert to understand.

We agree with the reviewer that the initial explanation of the mathematical modeling in **Fig. S7** and **S8** was very brief and difficult for non-specialist readers to follow. Our intent was to avoid distracting from the core biological findings, but we recognize the value of making this analysis more accessible. In the revised manuscript, we expanded the main text and the figure legends associated with this section to provide a clearer explanation of the model's structure, assumptions, and implications:

“Based on these results, we wondered whether consideration of global metabolic activity improves our understanding of T-cell turnover and subset dynamics. To this end, we developed mathematical models that provide a mechanistic framework to describe T-cell proliferation, differentiation and death in response to YFV vaccination – with or without metabolic activity (BPS) as a model parameter. We then tested the ability of different model assumptions in explaining the observed dynamics by adapting these models to the experimental data and comparing their deviation from the actual measurements (**Fig. S7-S8**). Assuming a linear differentiation pathway following (T_N (-like)- T_{SCM} - T_{CM} - T_{EM} - T_E), we found that including BPS of individual T-cell subsets significantly improved the quantification and identification of cellular dynamics compared to alternative approaches that assume time-constant rates for cell differentiation and turnover (Buchholz et al., 2013), or are based on Ki-67 levels. However, we also found that further, unknown subset-dependent factors likely influence cellular turnover. This is indicated by the best performing model evaluated, which allows for T_N (-like)/ T_{SCM} - and T_{CM}/T_E - and T_{EM} -subsets to differ in the scaling factor that regulates the influence of the measured subset-specific metabolic activity on the cellular turnover dynamics, in comparison to approaches that assume a common scaling factor for all subsets (**Fig. S7D-E**). Of note, these modeling results do not formally exclude the possibility that less differentiated subsets like T_N (-like) cells may directly differentiate into subsets like T_{EM} cells without a linear, stepwise differentiation program.”

Reviewer #2

(Remarks to the Author)

Schober and colleagues perform single cell studies to define the metabolic regulation of antigen-specific T cells in humans. By performing analyses on an interesting cohort of individuals vaccinated with the highly protective yellow fever vaccine, the authors longitudinally follow antigen-specific CD8+ T cells by using single cell RNA-seq and the multidimensional flow cytometry-based SCENITH protocol. They identify a differential metabolic regulation of CD8+ T cell subsets. Despite the high technological tour de force applied to precious human samples, the main message of the manuscript remains a bit unclear to the reviewer. While description of dynamics and changes following perturbations (in this case, vaccination) are important in human studies, the highly correlative nature of the results prevents the identification of important implications for long-lived T cell memory.

We appreciate the reviewer's recognition of our substantial technical effort to apply advanced single-cell approaches to precious human samples. We were surprised that the main message was perceived as unclear, given that both reviewers accurately described key details and conclusions of our study. In our effort to enhance clarity, already in the original version we included a graphical abstract and a conceptual model (formerly Fig. 5, now **Fig. 6**), which we hoped would aid in synthesizing the central findings and their implications. In our eyes, the revision process helped to increase the clarity of our manuscript even further, thanks to the insightful and constructive feedback given by both reviewers.

The most important message of our work is summarized in the title: "Metabolic quiescence precedes and maintains human antigen-specific T-cell memory *in vivo*". This refers to T_N(-like) cells showing a healthy-quiescent state throughout a physiological human T-cell response, which is associated with preferential maintenance of these stem-like cells over years to decades. Further key findings are highlighted in the abstract: "During the acute phase, T cells upregulated glycolysis to fuel anabolic needs of proliferation, but predominantly used oxidative phosphorylation for energy production, as assessed via protein translation rates. Central memory T cells were the most active subset, while effector cells underwent metabolic shut-down."

In the revised version of the manuscript, we now additionally summarize the main findings of our study early on and *en bloc* in the discussion section. While this introduces some redundancy (since we pick up individual findings in later discussion sections again when relevant), we hope that this further increases the clarity:

"Here, we harnessed protein translation to assess the metabolic activity and pathway dependence of human antigen-specific T cells after *in-vivo* immunization. Metabolic quiescence preceded and maintained human antigen-specific T-cell memory, with T_N-like cells showing a healthy-quiescent state throughout the immune response and forming the dominant memory population. While T_{CM} cells were the metabolically most active subset, T_{EM} and T_E cells showed an unhealthy-quiescent state. Energy production was most dependent on OXPHOS even during the acute phase of the immune response. Glycolytic dependence was most prominent in active T cells with an intermediate differentiation degree, suggesting a role for anabolism."

Our observations are based on the actual measurement of metabolic flux with subset (sometimes even single-cell) resolution, which had not been done before in mice or humans. In the human system – in which it is inherently more difficult to perform perturbations – such observational techniques are particularly important to obtain meaningful insights into medically relevant immunity. In mice, genetic knockouts are more feasible and allow specialized investigation, but such knockouts can also lead to secondary effects or feedback loops, which

can also make interpretation complicated. We view murine and human studies as complementary: mechanistic precision from mouse models paired with translational relevance from human systems. Together, they provide a more complete picture of the principles that govern long-lived T-cell memory. Of note, in this revision we not only validated our *ex-vivo* results in additional human and murine model systems (**Fig. S15-S17**), but also present several new *in-vitro* experiments using primary human T cells, which helped to investigate key aspects of our work in a more mechanistic manner (**Fig. S11-S13**). This confirmed that the association between T-cell differentiation status and metabolic activity which we observed after YFV vaccination was conserved across species and activation settings. We believe that our initial and these additional analyses, in conjunction with previous work by others, demonstrate the generalizability of our findings and their importance for our understanding of human T-cell memory formation.

In the revised version of the manuscript, we also adapted the final paragraph of the introduction to make some of these aspects clearer:

Example from introduction: “It has been known for more than five decades that T cells become metabolically active after *in-vitro* activation, with elevated rates of glycolysis, and that resting cells are quiescent, feeding on mitochondrial respiration (Chang et al., 2013; Pearce et al., 2009; Roos and Loos, 1973; van der Windt et al., 2012). Yet, recent mouse studies made clear that *in-vivo* activated T cells maintain a high dependence on OXPHOS even during rapid expansion in the acute phase of an immune response (Frisch et al., 2025; Ma et al., 2019). In mice, genetic knockouts allow mechanistic investigation more easily compared to studies in humans (Liikanen et al., 2021; Phan et al., 2016; Sukumar et al., 2013), but have led to conflicting observations on the interplay of metabolism and T-cell memory formation. T cell-specific knockout of Von-Hippel Lindau stabilizes HIF-1 α , elevates glycolysis and reduces OXPHOS. This leads to accelerated differentiation of T_{CM} cells into more differentiated T_{EM} cells, yet retains functional memory formation despite a decreased respiratory metabolism (Liikanen et al., 2021; Phan et al., 2016). Conversely, overexpression of the glycolytic enzyme phosphoglycerate mutase 1 impairs T-cell memory formation, while memory is enhanced by inhibition of glycolysis through 2-deoxy-D-glucose (2-DG) (Sukumar et al., 2013). Despite this important prior work, overall, the precise metabolic states of memory precursors *in vivo* require better definition. Population-level assessments fail to capture subset-specific metabolic dependencies, underscoring the need for direct measurements of antigen-specific T-cell metabolism at subset- or single-cell resolution throughout the immune response. This particularly applies to studies in humans.”

Major comments

1. There is a major concern on the use of some assays on cryopreserved cells. Although very long studies on human fresh samples are not doable for obvious reasons, the paper reports in multiple instances inconsistencies between cryopreserved and fresh samples (puromycin assay, caspase 3 assay). Moreover, it is general knowledge that CD62L expression is generally lost on cryopreserved samples (PMID: 12957403; PMID: 15286731). It might be partially recovered following overnight rest in medium. However, distribution among specific T cell subsets will be affected due to differential apoptosis of these cells. Therefore, definition of memory subsets on the basis of CD62L on cryopreserved cells is a possible matter of concern.

We appreciate the reviewer’s concern regarding the impact of cryopreservation on CD62L expression and subset definition. As noted, the use of cryopreserved samples was essential to perform technically demanding assays across a large cohort of donors and timepoints. Importantly, key analyses are shown both for cryopreserved and whole blood samples in the

supplementary figures (see, e.g., **Fig. S5, S6, S10, S14**). While we did expect to observe more apoptotic signaling in cryopreserved samples, metabolic activity as well as dependencies on OXPHOS and glycolysis were highly comparable between cryopreserved and whole blood samples.

We recognize that cryopreservation can affect surface marker expression, including CD62L, but took this into account during experimental design and analysis. While CCR7 would have been a valuable alternative marker for early differentiated T cells, its use was limited by staining conditions (optimal at 37°C), which are incompatible with key metabolic assays. We did observe some signs of CD62L shedding in our own data – mainly early after vaccination, but less so at later timepoints (**Fig. S1E-G**). Importantly, in our CITEseq dataset, signal intensity for CCR7 was relatively weak, while CD62L showed a robust signal (**Fig. S4A**). Because we aimed for maximal comparability between the flow cytometry and the CITEseq data, we decided to use CD62L, as it was reliably measurable across both platforms. Comparing CITEseq defined subsets with their transcriptional identity, we saw that cells that we classified as T_{EM} or T_E (i.e., based on CD62L negativity) were highly enriched for cells that corresponded to T_{EM} or T_E on a transcriptional level (**Fig. S4C**), arguing against a major hidden inclusion of stem-like cells that merely lost CD62L surface protein (note that all scRNAseq experiments were conducted with cryopreserved cells).

To further test the validity of CD62L as a marker for stem-like cells in an independent dataset of cryopreserved cells, we analyzed scRNAseq data with 130 CITEseq markers that we recently published (Kocher et al., 2024). We queried which surface protein markers were through their presence (positive marker) or absence (negative marker) best positive or negative predictors of cells with a T_N (-like) transcriptional signature. This yielded expression of CD45RA and CD62L as the top positive markers next to absence of CD95 as the top negative marker (**Fig. S4D**). To explain this analysis in more detail, the expression (presence) of CD62L predicts positively cells to have a T_N (-like) phenotype with about 50% likelihood (positive predictive value of about 0.5). The absence of CD62L suggests with roughly 97% likelihood that a cell does not have a T_N (-like) phenotype. For the negative markers, a positive predictive value of about 0.55 for CD95 implies that – if a cell is CD95-positive – then it will not be T_N (-like) with 55% likelihood. Conversely, a negative predictive value for CD95 of 0.98 shows that 98% of T_N (-like) cells are CD95-negative.

The absence of CCR7 among surface markers with high predictive power should be interpreted with caution since our staining protocol (at 4°C) prior to scRNAseq may not have been ideally compatible with CCR7 staining. However, these two independent datasets still demonstrate that CD62L protein surface expression is a biologically meaningful marker for early-differentiated T-cell subsets. In the revised version of the manuscript, we elaborate on these points in the results section:

“While CD62L shedding is a known phenomenon, our data suggest it primarily occurred early after vaccination, with minimal impact at later timepoints (**Fig. S1E-G**). In our CITEseq data, CCR7 signal was weak, while CD62L was robustly detected. To ensure comparability with flow cytometry, we used CD62L as a stemness marker since it was reliably detected in flow cytometry and CITEseq. T_{EM} or T_E classification based on CD62L negativity aligned with transcriptional identities (**Fig. S4C**). Moreover, in an independent scRNAseq dataset (Kocher et al., 2024) with 130 CITEseq markers, CD45RA and CD62L emerged as top predictors of naïve-like transcriptional profiles (**Fig. S4D**). These data support CD62L surface protein as a meaningful marker for T cells with early differentiation profiles.”

2. As I perceive it, the main message the authors want to convey is that central memory T cells (TCM) are the most active phenotypic subset during response to yellow fever vaccine, while effector cells undergo metabolic shutdown. The implication of these differences in relation to long-lived T cell memory are unclear. In which way high basal protein synthesis should impact proliferation of TCM (page 5)? Could also the opposite be true? Results appear highly correlative and the implications of the findings are unclear in the absence of specific mechanistic investigations. These could be done on polyclonal memory T cells due to the paucity of antigen-specific subsets.

We appreciate the reviewer's careful reading and agree that one of the central findings of our study is that T_{CM} cells are the most metabolically active subset during the acute phase of the YFV response, whereas T_E cells exhibit signs of metabolic shutdown. We would also like to emphasize through both our manuscript title and abstract that another key insight from our work was the identification of T_N(-like) antigen-specific cells as metabolically quiescent and long-lived. These cells, despite being weakly activated, became the dominant population at memory timepoints. This finding aligns with broader principles in cellular biology, where quiescence is often associated with longevity and self-renewal potential. Our study contributed new evidence specific to human memory T-cell subsets in the context of a physiologically relevant *in-vivo* antigen exposure.

We interpret the relationship between basal protein synthesis (BPS) and proliferation as mutually reinforcing processes: high biosynthetic activity supports proliferation, and conversely, proliferating cells may upregulate translation machinery. Establishing directionality in this context is inherently difficult. However, previous *in-vitro* experiments indicate that an initial translational burst precedes the first division of T cells upon activation, suggesting that enhanced protein translation might prepare T cells for proliferation (Marchingo and Cantrell, 2022).

For the revised version of the manuscript, we performed several new *in-vitro* analyses, which confirmed this concept. On the one hand, we extended our "*in-vitro* SCENITH" set-up by additional timepoints (**Fig. S11**). Here, polyclonal primary human T cells were stimulated by aCD3/aCD28 with low-dose IL-2 for 24h, 48h or 72h, and SCENITH was performed at each timepoint prior to flow cytometric analysis. On the other hand, in a further set of experiment, we stimulated polyclonal primary human T cells in the same manner, but applied inhibitor treatments (oligomycin, 2-DG, harringtonine or control treatment) in the initial 24h (**Fig. S12-S13**). This enabled us to investigate the influence of early metabolic processes on subsequent proliferation, differentiation and activation. We did this for unsorted cells and for cells that were initially flow cytometrically sorted for different phenotypic subsets.

This additional investigation clearly demonstrated that BPS and activation marker upregulation were highest at 24h, and thereby preceded proliferation which was most prominent only later at 72h (**Fig. S11B-E, Fig. S12D**). Furthermore, initial translation inhibition through harringtonine and OXPHOS inhibition through oligomycin completely prevented later proliferation (**Fig. S12F,H, Fig. S13B**). Notably, inhibition of glycolysis only affected cell counts upon stimulation, but not in the unstimulated condition (**Fig. S12H, Fig. S13B**). This underlines that inhibition of glycolysis showed the most prominent effect on stimulated proliferative cells, which happened to be T_{CM} cells *in vitro* (**Fig. S11C, Fig. S12E**), in accordance with our *in-vivo* results.

Overall, we consider our new *in-vitro* analyses (**Fig. S11-S13**) to be helpful in providing more mechanistic insights, complementing our *ex-vivo* findings. The results showed that protein synthesis in general was essential for T-cell proliferation and activation and that all T cells were strongly dependent on OXPHOS with regard to (proliferative) activity, but that simultaneously most glycolytic dependence was detectable in highly proliferative cells, such as T_{CM} are *in vivo*.

3. In some parts, results do not appear completely novel over previous literature. Figure 1 nicely confirms the dynamics of antigen-specific immune responses in response to vaccination, with some minor differences (e.g., definition of subsets, but please also see point 1; PMID: 18468462; PMID: 25855494). In Fig. 4, the authors report that tendency to apoptosis (cleaved caspase 3 in viability dye-negative cells) increases with progressive differentiation of antigen-specific T cells. This is somewhat expected and known from previous studies (PMID: 14625547; PMID: 23281401).

We thank the reviewer for this comment. We fully acknowledge the important contributions of prior studies in characterizing T-cell subset dynamics, differentiation, and susceptibility to apoptosis following antigen exposure, and now also cite the studies the reviewer mentioned that we had not cited before (Kaech et al., 2003; Lugli et al., 2013; Miller et al., 2008) to make clear that we are not claiming novelty at inappropriate occasions.

Example from the discussion: “While T_{CM} cells were most active, we detected sub-populations of T_{EM} and T_E cells with reduced metabolic activity ($puro^{lo}$) and a predisposition for apoptosis next to $puro^{hi}$ T_{EM} and T_E cells. In line with this, more differentiated cells are prone to contraction (Kaech et al., 2003; Lugli et al., 2013) and have shorter half-lives *in vivo*, although they can be maintained for a remarkably long time (Dolfi et al., 2008; Garrod et al., 2012; Kaech et al., 2002).”

The primary gap our study did address is the lack of direct measurements of metabolic flux in *in-vivo* activated antigen-specific T cells with subset-level resolution in both humans and mice. While phenotypic kinetics and increased apoptosis in more differentiated T-cell subsets had been reported before, our findings are novel in how they correlate apoptosis susceptibility with direct functional readouts of metabolic activity, including protein synthesis rates – all in a standardized human T-cell response following vaccination with subset or even single-cell resolution. Thus, although the apoptosis trends were consistent with earlier work, they reinforced and integrated with our new metabolic data, highlighting coordinated regulation of survival, metabolic state, and differentiation in humans *in vivo*.

4. Fig. 4C relies on very few events per single subset, therefore quantification of cells in puromycin/caspase 3 gates is highly subjected to measurement errors. E.g., there is about 10 T_{CM} cells at 1 year, thus every cell contributes about 10% to the quantification, thus certainly skewing results and statistics in Fig. 4D.

We appreciate the reviewer’s attention to the limitations of low cell counts in some phenotypic subsets, particularly in assays such as puromycin incorporation and cleaved Caspase-3 staining. Recognizing this issue, we applied a strict cut-off across all analyses in our manuscript: only subsets with at least 10 cells were included for downstream quantifications. This can, in some instances, result in final populations with even less cells. While we agree that a subset with few events can introduce variability if considered in isolation, the conclusions in the former Fig. 4D (now **Fig. 5D**) as well as in all other key analyses were based on aggregate data from a substantial number of YFV donors across timepoints – ranging from 4 to 38 donors per group. Specifically, as detailed in **Table S2**, the d14 timepoint included 38 donors, and the 1yr timepoint included 20 donors. This broad sampling enabled us to observe highly consistent trends with relatively small error margins, mitigating the risk of skewed interpretation due to low event counts in isolated cases.

We can showcase this with the reviewer’s example of T_{CM} cells from **Fig. 5D**. Because of our cut-offs, the T_{CM} analyses were restricted to 35 out of 38 (d14) or 10 out of 20 (1yr) donors.

However, this remaining number of biological replicates still ensured robust detection of phenotypes, as can be seen by statistically significant differences in biologically meaningful comparisons and the visualized error margins (**Fig. 5D**). For the reviewer's information, in the following table we additionally report the mean numbers of epitope-specific T_{CM} cells for each metabolic subpopulation, along with the corresponding SEM values, which form the basis for the results displayed in **Fig. 5D**.

TCM cells	Timepoint	Mean (cell counts)	SEM (cell counts)	n (donors)
Puro⁻/clCasp3⁻	d14	22.9	4.0	35
	1yr	67.4	7.3	10
Puro⁺/clCasp3⁻	d14	40.8	3.7	35
	1yr	28.5	7.7	10
Puro⁺/clCasp3⁺	d14	23.2	3.7	35
	1yr	2.1	1.5	10
Puro⁻/clCasp3⁺	d14	12.5	2.8	35
	1yr	1.8	1.2	10

Table R1: Exemplary cell counts for metabolic subpopulations of T_{CM} cells in Fig. 5D.

5. Metabolic analysis would benefit from integration/correlation with scRNA-seq data and transcriptional activity/identity. The authors have a very precious dataset that would require more in depth analyses.

We thank both reviewers for raising this point. We had indeed considered the potential of our scRNAseq dataset for metabolic profiling but initially decided to focus on SCENITH-derived measurements of metabolic flux, which we view as more functionally informative. RNA expression levels alone do not always reflect protein levels and overall pathway activity.

However, we do agree that incorporating transcriptomic analysis of metabolic signatures enhances the manuscript and provides a more balanced and comprehensive assessment. In the revised version, we therefore integrated in-depth analyses of our scRNAseq data to assess glycolysis and OXPHOS pathway transcription, as well as broader metabolic and immunological programs (new **Fig. 3**). We thereby explored the use of advanced computational tools such as Vision and COMPASS, which use weighted gene scores and thereby counteract some of the concerns phrased above. Using Vision (DeTomaso et al., 2019), we re-performed transcriptional pathway analyses that we had done before using decoupleR (Badia-I-Mompel et al., 2022) and also performed new analyses. Vision often yielded similar results as decoupleR, but in some instances more consistently matched our flow cytometry-based results.

To obtain a comprehensive overview beyond our initial analyses (**Fig. 3A**), we investigated the mean scores for individual pathways (see rows in **Fig. 3B** and **Fig. S9C**), and plotted them for A2/NS4B-specific T cells as row-normalized z-scores per timepoint, per selected Leiden cluster or per CITEseq phenotype. This demonstrated that quiescence was downregulated in the acute phase of the immune response, in which OXPHOS and glycolysis were prominent. Absolute pathway scores (see horizontal bars on the right in **Fig. 3B** and **Fig. S9C**) are based

on overall transcript levels per gene in a given pathway. Assuming that these scores reflect the biological significance of a pathway, OXPHOS signatures were relatively more prominent than glycolysis signatures even at acute timepoints. Generally, the pathway scores correlate both with absolute metabolic activity and relative dependence on the pathway, the two of which cannot be distinguished. Note that, in comparison, SCENITH measures absolute activity through BPS and relative dependencies on specific pathways separately.

In terms of phenotypic subsets, quiescence signatures were most prominent in clusters and CITEseq populations corresponding to T_N (-like) cells or non-cycling T_{CM} cells (**Fig. 3B**). In cycling cells, OXPHOS and glycolysis were both upregulated compared to other Leiden clusters, although the absolute pathway scores suggest a larger role for OXPHOS as explained above. Please note that cycling cells represented the majority of cells at d14 and mostly had a T_{CM} and T_{EM} CITEseq phenotype (**Fig. 1I, S4B**). Non-cycling T_{CM1} and T_{CM2} cells were relatively more active in glycolysis, while OXPHOS was more engaged by T_N (-like) cells (**Fig. 3B**). This was confirmed on the CITEseq level. For the core pathways OXPHOS, glycolysis and T-cell proliferation, we also visualized longitudinal timelines, UMAPs and violin plots (**Fig. 3C-E**). These data indicated that, time-wise, OXPHOS genes were increased in $A2/NS4B^+$ cells compared to $A2/NS4B^-$ cells in cycling cluster at d14, but similarly low again at one year (**Fig. 3C**). Smaller (on an absolute, but not relative level) yet consistent differences were noticeable for the glycolysis score. Both metabolic scores showed an earlier peak (d11) than the proliferation score (d14), which mirrored our flow cytometric data with BPS peaking before activation markers or Ki-67 (**Fig. S5C, S6F, S6K**). On a cluster level, OXPHOS genes were highest in the cycling cluster, also high in naïve-like clusters and lowest in the more differentiated T_{EM} and T_E clusters (**Fig. 3D-E**). The glycolysis score showed similar patterns, but was elevated in non-cycling T_{CM} cells and not in T_N (-like) clusters.

Our new analyses yielded several additional findings. For example, pathways for apoptosis and base excision repair followed similar longitudinal kinetics and were both expectedly upregulated in cycling cells, but showed key differences in the distribution across transcriptional clusters and CITEseq subsets (**Fig. 3B**). Apoptosis was upregulated in T_{EM} and T_E clusters with IFN signatures, indicating contraction in these populations during the “cooling phase” of the immune response. In contrast, base excision repair was upregulated in the T_N (-like) clusters, indicating that stem-like cells are transcriptionally prepared for genomic surveillance without showing elevated levels of γ H2Ax protein in the absence of DNA damage (compare to **Fig. 5F**). Finally, we also explored additional metabolic pathways for patterns with regards to longitudinal dynamics or subset distribution. This led to novel observations, for example an upregulation of linoleic acid metabolism (Nava Lauson et al., 2023) specifically in the memory phase, while arachidonic acid signatures (Bibby et al., 2022) were prominent in the early acute (d7) as well as in the memory phase. We report these data as an outlook and resource for future investigation (**Fig. S9C**).

Our flow cytometry data indicated that T_{CM} cells consisted of a proliferative and metabolically active subpopulation in parallel to resting T_{CM} cells (see, e.g., **Fig. 2C** or **Fig. 5C**). In the scRNAseq data, we found that, at d14, the majority of CITEseq-defined T_{CM} cells were located within the cycling cluster (**Fig. S9D**), which also comprised CITEseq-defined T_{EM} cells (**Fig. S4B**). Importantly, comparing the enriched/depleted pathways in T_{CM} and T_{EM} cells within the cycling cluster revealed that cycling T_{CM} cells were more active and proliferative with a more pronounced transcriptional OXPHOS signature than cycling T_{EM} cells (**Fig. S9F**). This recapitulated our flow cytometry-derived metabolic data.

In summary, phenotypic subsets displayed distinct metabolic transcriptional programs during the course of a human T-cell response. While quiescence was associated with stem-like T cells and thus enriched at d0 and at late memory timepoints, metabolic pathways peaked at d11 in cycling cells and suggested a high relevance for OXPHOS during this acute phase.

Minor concerns

1. At page 7, second paragraph, the authors state that “these *ex vivo* results differ from *in vitro* conditions in which stimulated T cells are more dependent on glycolysis and less in mitochondrial ATP production”. It should be noted that activated T cells *in vitro* show a metabolic switch from OXPHOS to glycolysis, nevertheless they keep relying on OXPHOS at high rates (e.g, PMID: 22206904 Fig. 6 or PMID: 31747582 Fig. 1).

We thank the reviewer for this helpful clarification. We agree that *in-vitro* activated T cells continue to rely significantly on OXPHOS, even after undergoing a glycolytic switch. For the revised version of the manuscript, we extended our initial SCENITH analyses of *in-vitro* stimulated T cells. As shown in **Fig. S11**, our initial and extended data demonstrate that *in-vitro* stimulated T cells exhibited approximately 20-40% dependence on OXPHOS across various timepoints (24h, 48h and 72h after stimulation), while glycolytic dependence decreased from about 40% at 24h to < 10% at 72h. At the acute *in-vivo* timepoint (d14 post-vaccination; **Fig. 4B**), we observed an OXPHOS dependency (47%) and glycolytic contribution (13%) that best matched results from 72h post *in-vitro* stimulation.

Our initial intent (when we had only analyzed the 24h timepoint *in vitro*) was to highlight these quantitative differences between *in-vitro* and *in-vivo* conditions and not to suggest that OXPHOS is dispensable *in vitro*. We now revised the relevant section of the manuscript to report on commonalities and differences between the *in-vitro* and *in-vivo* results in a balanced manner by deleting the statement that *in-vitro* stimulated cells depended less on mitochondrial respiration. We also incorporated the cited references (Ma et al., 2019; van der Windt et al., 2012) to place our results in the appropriate context:

“Notably, these *ex-vivo* results differ from *in-vitro* conditions in which stimulated T cells are more dependent on glycolysis (Frisch et al., 2025; Ma et al., 2019; Sukumar et al., 2013; van der Windt et al., 2012). SCENITH on *in-vitro* activated cells revealed that glycolytic dependence was especially high (around 40%) 24h after polyclonal stimulation, when BPS was also highest (**Fig. S11**). At this early timepoint, both glycolytic dependence and BPS were particularly high in T_{SCM} and T_{CM} cells, while mitochondrial dependence was highest in T_N(-like) cells. Interestingly, the metabolic parameters of *in-vitro* activated T cells after 72h best matched the results we obtained at the acute (d14) timepoint post YFV vaccination – with higher reliance on OXPHOS compared to glycolysis – albeit some differences between *in-vitro* and *in-vivo* conditions prevailed regarding the phenotypic subset level.”

References

- Akondy, R.S., Fitch, M., Edupuganti, S., Yang, S., Kissick, H.T., Li, K.W., Youngblood, B.A., Abdelsamed, H.A., McGuire, D.J., Cohen, K.W., et al. (2017). Origin and differentiation of human memory CD8 T cells after vaccination. *Nature* *552*, 362–367.
- Badia-I-Mompel, P., Vélez Santiago, J., Braunger, J., Geiss, C., Dimitrov, D., Müller-Dott, S., Taus, P., Dugourd, A., Holland, C.H., Ramirez Flores, R.O., et al. (2022). decoupleR: ensemble of computational methods to infer biological activities from omics data. *Bioinforma. Adv.* *2*, 1–3.
- Bibby, J.A., Agarwal, D., Freiwald, T., Kunz, N., Merle, N.S., West, E.E., Singh, P., Larochelle, A., Chinian, F., Mukherjee, S., et al. (2022). Systematic single-cell pathway analysis to characterize early T cell activation. *Cell Rep.* *41*, 111697.
- Bresser, K., Kok, L., Swain, A.C., King, L.A., Jacobs, L., Weber, T.S., Perié, L., Duffy, K.R., de Boer, R.J., Scheeren, F.A., et al. (2022). Replicative history marks transcriptional and functional disparity in the CD8+ T cell memory pool. *Nat. Immunol.* *23*, 791–801.
- Buchholz, V.R., Flossdorf, M., Hensel, I., Kretschmer, L., Weissbrich, B., Gräf, P., Verschoor, A., Schiemann, M., Höfer, T., and Busch, D.H. (2013). Disparate individual fates compose robust CD8+ T cell immunity. *Science* *340*, 630–635.
- Buchholz, V.R., Schumacher, T.N.M., and Busch, D.H. (2016). T Cell Fate at the Single-Cell Level. *Annu. Rev. Immunol.* *34*, 65–92.
- Buck, M.D.D., O'Sullivan, D., Klein Geltink, R.I.I., Curtis, J.D.D., Chang, C.H., Sanin, D.E.E., Qiu, J., Kretz, O., Braas, D., van der Windt, G.J.J.W., et al. (2016). Mitochondrial Dynamics Controls T Cell Fate through Metabolic Programming. *Cell* *166*, 63–76.
- Chang, C.-H., Curtis, J.D., Maggi, L.B., Faubert, B., Villarino, A. V., O'Sullivan, D., Huang, S.C.-C., van der Windt, G.J.W., Blagih, J., Qiu, J., et al. (2013). Posttranscriptional Control of T Cell Effector Function by Aerobic Glycolysis. *Cell* *153*, 1239–1251.
- DeTomaso, D., Jones, M.G., Subramaniam, M., Ashuach, T., Ye, C.J., and Yosef, N. (2019). Functional interpretation of single cell similarity maps. *Nat. Commun.* *10*.
- Dolfi, D. V., Boesteanu, A.C., Petrovas, C., Xia, D., Butz, E.A., and Katsikis, P.D. (2008). Late Signals from CD27 Prevent Fas-Dependent Apoptosis of Primary CD8+ T Cells. *J. Immunol.* *180*, 2912–2921.
- Frisch, A.T., Wang, Y., Xie, B., Yang, A., Ford, B.R., Joshi, S., Kedziora, K.M., Peralta, R., Wilfahrt, D., Mullett, S.J., et al. (2025). Redirecting glucose flux during in vitro expansion generates epigenetically and metabolically superior T cells for cancer immunotherapy. *Cell Metab.* 1–16.
- Fuertes Marraco, S.A., Soneson, C., Cagnon, L., Gannon, P.O., Allard, M., Maillard, S.A., Montandon, N., Rufer, N., Waldvogel, S., Delorenzi, M., et al. (2015). Long-lasting stem cell-like memory CD8+ T cells with a naive-like profile upon yellow fever vaccination. *Sci. Transl. Med.* *7*, 282ra48-282ra48.
- Garrod, K.R., Moreau, H.D., Garcia, Z., Lemaître, F., Bouvier, I., Albert, M.L., and Bousso, P. (2012). Dissecting T Cell Contraction In Vivo Using a Genetically Encoded Reporter of Apoptosis. *Cell Rep.* *2*, 1438–1447.
- Gattinoni, L., Lugli, E., Ji, Y., Pos, Z., Paulos, C.M., Quigley, M.F., Almeida, J.R., Gostick, E., Yu, Z., Carpenito, C., et al. (2011). A human memory T cell subset with stem cell-like properties. *Nat. Med.* *17*, 1290–1297.
- Geginat, J., Lanzavecchia, A., and Sallusto, F. (2003). Proliferation and differentiation potential of human CD8+ memory T-cell subsets in response to antigen or homeostatic cytokines. *Blood* *101*, 4260–4266.
- Gubser, P.M., Bantug, G.R., Razik, L., Fischer, M., Dimeloe, S., Hoenger, G., Durovic, B., Jauch, A., and Hess, C. (2013). Rapid effector function of memory CD8+ T cells requires an immediate-early glycolytic switch. *Nat. Immunol.* *14*, 1064–1072.
- Kaech, S.M., Hemby, S., Kersh, E., and Ahmed, R. (2002). Molecular and functional profiling of memory CD8 T cell differentiation. *Cell* *111*, 837–851.
- Kaech, S.M., Tan, J.T., Wherry, E.J., Konieczny, B.T., Surh, C.D., and Ahmed, R. (2003). Selective expression of the interleukin 7 receptor identifies effector CD8 T cells that give rise to long-lived memory cells. *Nat. Immunol.* *4*, 1191–1198.
- Klebanoff, C.A., Scott, C.D., Leonardi, A.J., Yamamoto, T.N., Cruz, A.C., Ouyang, C., Ramaswamy, M., Roychoudhuri, R., Ji, Y., Eil, R.L., et al. (2016). Memory T cell-driven differentiation of naive cells impairs adoptive immunotherapy. *J. Clin. Invest.* *126*, 318–334.
- Kocher, K., Drost, F., Tesfaye, A.M., Moosmann, C., Schuelein, C., Grotz, M., D'Ippolito, E., Graw, F., Spriewald, B., Busch, D.H., et al. (2024). Quality of vaccination-induced T cell responses is conveyed by polyclonality and high, but not maximum, antigen receptor avidity. *BioRxiv*.
- Levine, L.S., Hiam-Galvez, K.J., Marquez, D.M., Tenvooren, I., Madden, M.Z., Contreras, D.C., Dahunsi, D.O., Irish, J.M., Oluwole, O.O., Rathmell, J.C., et al. (2021). Single-cell analysis by mass cytometry reveals metabolic

states of early-activated CD8⁺ T cells during the primary immune response. *Immunity* 54, 829-844.e5.

Li, J., Zaslavsky, M., Su, Y., Guo, J., Sikora, M.J., van Unen, V., Christophersen, A., Chiou, S., Chen, L., Li, J., et al. (2022). KIR⁺CD8⁺ T cells suppress pathogenic T cells and are active in autoimmune diseases and COVID-19. *Science* 376, eabi9591.

Liikanen, I., Lauhan, C., Quon, S., Omilusik, K., Phan, A.T., Bartrolí, L.B., Ferry, A., Goulding, J., Chen, J., Scott-Browne, J.P., et al. (2021). Hypoxia-inducible factor activity promotes antitumor effector function and tissue residency by CD8⁺ T cells. *J. Clin. Invest.* 131.

Lugli, E., Dominguez, M.H., Gattinoni, L., Chattopadhyay, P.K., Bolton, D.L., Song, K., Klatt, N.R., Brenchley, J.M., Vaccari, M., Gostick, E., et al. (2013). Superior T memory stem cell persistence supports long-lived T cell memory. *J. Clin. Invest.* 123, 594–599.

Ma, E.H., Verway, M.J., Johnson, R.M., Roy, D.G., Steadman, M., Hayes, S., Williams, K.S., Sheldon, R.D., Samborska, B., Kosinski, P.A., et al. (2019). Metabolic Profiling Using Stable Isotope Tracing Reveals Distinct Patterns of Glucose Utilization by Physiologically Activated CD8⁺ T Cells. *Immunity* 51, 856-870.e5.

Ma, R., Ji, T., Zhang, H., Dong, W., Chen, X., Xu, P., Chen, D., Liang, X., Yin, X., Liu, Y., et al. (2018). A Pck1-directed glycogen metabolic program regulates formation and maintenance of memory CD8⁺ T cells. *Nat. Cell Biol.* 20, 21–27.

Marchingo, J.M., and Cantrell, D.A. (2022). Protein synthesis, degradation, and energy metabolism in T cell immunity. *Cell. Mol. Immunol.* 19, 303–315.

Miller, J.D., van der Most, R.G., Akondy, R.S., Glidewell, J.T., Albott, S., Masopust, D., Murali-Krishna, K., Mahar, P.L., Edupuganti, S., Lalor, S., et al. (2008). Human Effector and Memory CD8⁺ T Cell Responses to Smallpox and Yellow Fever Vaccines. *Immunity* 28, 710–722.

Nava Lauson, C.B., Tiberti, S., Corsetto, P.A., Conte, F., Tyagi, P., Machwirth, M., Ebert, S., Loffreda, A., Scheller, L., Sheta, D., et al. (2023). Linoleic acid potentiates CD8⁺ T cell metabolic fitness and antitumor immunity. *Cell Metab.* 35, 633-650.e9.

Nicoli, F., Papagno, L., Frere, J.J., Cabral-Piccin, M.P., Clave, E., Gostick, E., Toubert, A., Price, D.A., Caputo, A., and Appay, V. (2018). Naïve CD8⁺ t-cells engage a versatile metabolic program upon activation in humans and differ energetically from memory CD8⁺ T-cells. *Front. Immunol.* 9, 1–12.

Pearce, E.L., Walsh, M.C., Cejas, P.J., Harms, G.M., Shen, H., Wang, L.S., Jones, R.G., and Choi, Y. (2009). Enhancing CD8 T-cell memory by modulating fatty acid metabolism. *Nature* 460, 103–107.

Phan, A.T., Doedens, A.L., Palazon, A., Tyrakis, P.A., Cheung, K.P., Johnson, R.S., and Goldrath, A.W. (2016). Constitutive Glycolytic Metabolism Supports CD8⁺ T Cell Effector Memory Differentiation during Viral Infection. *Immunity* 45, 1024–1037.

Roos, D., and Loos, J.A. (1973). Changes in the carbohydrate metabolism of mitogenically stimulated human peripheral lymphocytes. II. Relative importance of glycolysis and oxidative phosphorylation on phytohaemagglutinin stimulation. *Exp. Cell Res.* 77, 127–135.

Sukumar, M., Liu, J., Ji, Y., Subramanian, M., Crompton, J.G., Yu, Z., Roychoudhuri, R., Palmer, D.C., Muranski, P., Karoly, E.D., et al. (2013). Inhibiting glycolytic metabolism enhances CD8⁺ T cell memory and antitumor function. *J. Clin. Invest.* 123, 4479–4488.

van der Windt, G.J.W., and Pearce, E.L. (2012). Metabolic switching and fuel choice during T-cell differentiation and memory development. *Immunol. Rev.* 249, 27–42.

van der Windt, G.J.W., Everts, B., Chang, C.H., Curtis, J.D., Freitas, T.C., Amiel, E., Pearce, E.J., and Pearce, E.L. (2012). Mitochondrial Respiratory Capacity Is a Critical Regulator of CD8⁺ T Cell Memory Development. *Immunity* 36, 68–78.

Wolf, T., Jin, W., Zoppi, G., Vogel, I.A., Akhmedov, M., Bleck, C.K.E., Beltraminelli, T., Rieckmann, J.C., Ramirez, N.J., Benevento, M., et al. (2020). Dynamics in protein translation sustaining T cell preparedness. *Nat. Immunol.* 21, 927–937.

Zhang, H., Liu, J., Yang, Z., Zeng, L., Wei, K., Zhu, L., Tang, L., Wang, D., Zhou, Y., Lv, J., et al. (2022). TCR activation directly stimulates PYGB-dependent glycogenolysis to fuel the early recall response in CD8⁺ memory T cells. *Mol. Cell* 82, 3077-3088.e6.

Zwijnenburg, A.J., Pokharel, J., Varnaité, R., Zheng, W., Hoffer, E., Shryki, I., Comet, N.R., Ehrström, M., Gredmark-Russ, S., Eidsmo, L., et al. (2023). Graded expression of the chemokine receptor CX3CR1 marks differentiation states of human and murine T cells and enables cross-species interpretation. *Immunity* 1–20.

POINT-BY-POINT RESPONSE TO REVIEWS

Frischholz, Schuster, Grotz et al. (NI-A40360A, "Metabolic quiescence of naive-like memory T cells precedes and maintains antigen-specific T-cell memory in vivo")

(Reviewer comments in *italic*, responses in non-italic and red)

Overview of main manuscript changes after second revision

- Updated title to: "Metabolic quiescence of naive-like memory T cells precedes and maintains antigen-specific T-cell memory"
- Revised terminology and clarified definition of T_N(-like) cells, now consistently referred to as naïve (T_N) or naïve-like memory T cells (T_{NM})
- Removed the term "metabolic flux" when describing SCENITH analyses, to accurately reflect the assay's readout.
- Editorial revisions: Relabeled "Supplementary Figures" into "Extended Data Figures" and "Supplementary Figures".

Reviewer #1

(Remarks to the Author)

In this manuscript, Frischholz et al. have performed multiple new experiments and analyses of human SARS-CoV2 vaccination and mouse LCMV infection models, which have strengthened the overall conclusions of the study. Importantly, they also performed new experiments and identified basal protein synthesis and OXPHOS as key regulators of the stem-like state of naïve T cell (T_N)-like cells in vitro, providing new mechanistic insight for how metabolism regulates the quiescence and self-renewal of this unique cell population that correlates with long-term protective immunity after yellow fever vaccination. They also performed additional analyses on their CITE-seq dataset. Overall, the authors have addressed my previous concerns, and I support its publication.

We are glad we could address all points raised and thank the reviewer for supporting the publication.

Reviewer #3

(Remarks to the Author):

Assessment of author rebuttal:

Reviewer 1

1a. The authors sufficiently responded to including an additional human antigen-specific system by adding past studies on SARS-CoV-2 antigen-specific T cells (Figure S15 and S16).

1b. What the reviewer requested was not technically feasible since memory T cells at later time points are all metabolically quiescent.

1c. The authors included additional timepoints for in vitro analysis of metabolic phenotypes (Figure S11).

2. The authors added an additional mouse infection model (LCMV Armstrong) and assessed virus specific responses using P14 TCR transgenic T cells at early (day 6 and day 10) and later (day 30) time points (Figure S17). These experiments are important, since they demonstrated key differences between mouse and human T cell responses following in vivo challenge.

3. In response to the question of whether Tn(-like) cells are more functional after antigen recall, the authors indicated that this was already addressed in a previous publication (Fuentes Marraco, 2015). However, the authors proceeded to examine the responses of polyclonal T cells following in vitro stimulation of PBMCs with anti-CD3/anti-CD28/IL-2 using multiple time points (24, 48, 72 hours) and assessing CD8⁺ T cell subsets before and after stimulation (Figure S11), following short-term metabolic perturbation of anti-CD3/anti-CD28/IL-2 stimulated PBMCs (Figure S12), and following CD3/anti-CD28/IL-2 stimulation of sorted CD8⁺ T cell "memory" subsets (Figure S13). Although these experiments provided valuable insights into the in vitro differentiation dynamics and metabolic remodeling of CD8⁺ T cells subpopulations following polyclonal stimulation, the interpretation of these findings in the context of Tn(-like) cell biology was overstated. The populations classified as Tn(-like) in these studies were, in fact, composed predominantly of bona fide naïve T cells. Therefore, the observed Tn(-like) responses most likely represent authentic naïve T cell (Tn) responses. Can the authors comment on this?

4. The ability of Tn(-like) memory cells to undergo IL-7/IL-15 dependent homeostatic proliferation was not addressed by the authors but response to IL-15 stimulation was assessed in a previous publication.

5. The authors used a stepwise differentiation mathematical model, which provided a predictive framework that agreed with their experimental observations (Figure S7 and S8). However, they do acknowledge the limitations of this approach and that experimentally testing this model in humans is extremely challenging.

6a. The authors performed additional analysis on metabolic and immune pathways on their CITE-seq datasets. They added transcriptome heatmaps comparing: (i) time after vaccination, (ii) Leiden clusters from scRNA-seq datasets, (iii) CITE-seq. These re-analyses demonstrated that Tn(-like) cells possessed largely similar transcripts between scRNA and CITE-seq datasets (Figure 2b). The authors also further analyzed transcripts of antigen specific T cells in CM, EM, E subsets (Figure S9A). Their conclusions were not altered following reanalysis.

6b. Metabolic pathway activities were re-evaluated and visualized using row-normalized z-score plots. This analysis revealed that oxidative phosphorylation (OxPhos) signatures were consistently elevated across all examined time points (Figure 3 and S9). Furthermore, the authors demonstrated that quiescence-associated transcriptional programs were enriched in Tn(-like) and Tcm cell populations. This re-analysis further supported the authors original conclusion that quiescence and Oxphos are key signatures of Tn(-like) memory cells.

6c. The issue of discriminating Tn(-like) and Tcm subclusters were also addressed by the authors.

6d. The authors employed two additional bioinformatic approaches that were suggested by the reviewer to further characterize transcriptional metabolic profiles, namely Vision and COMPASS. As stated by the authors, Vision yielded similar results when compared to the bioinformatics tool they originally used, decoupleR.

We thank the reviewer for evaluating whether the concerns originally raised by reviewer 1 were now sufficiently resolved. We appreciate the reviewer's assessment that our revisions adequately addressed all previously raised points.

We agree with the reviewer's interpretation regarding point 3. In the context of the polyclonal stimulation experiments (**Extended Data Fig. 5-7**; previously Fig. S11–S13), CD95⁻CD45RA⁺CD62L⁺ cells indeed should predominantly represent *bona fide* naïve (antigen-inexperienced) CD8⁺ T cells. We have now made this point more explicit in the revised manuscript to avoid potential over-interpretation and to clearly distinguish this setting from the naïve-like populations described in antigen-experienced contexts by others before and by us in this manuscript. In addition, we have updated the nomenclature to naïve-like memory T cells (T_{NM}) throughout the manuscript to make the distinction between antigen-inexperienced T_N and antigen-experienced T_{NM} cells clearer. We believe this refinement of nomenclature has strengthened the transparency and interpretability of the manuscript without changing the main conclusions.

Reviewer 2:

1. The reviewer expressed concerns on potential differences in experimental results between cryopreserved and fresh samples, in particular CD62L. The authors validated the appropriate use of the marker for identifying T_n(-like) cells by using scRNA-seq datasets from a previous publication (Figure S4D).

2. The authors clarified that the key finding in this manuscript is the metabolic characterization of T_n(-like) memory T cells. I think this is a major problem of the manuscript. The authors do not give enough background on what T_n(-like) memory cells are and their relevance in identified memory T cell subsets.

3. The reviewer raised concerns regarding the novelty of the findings presented in this manuscript. The authors responded by stating that combining metabolic readouts, such as SCENITH, with other modalities, further characterized a distinct CD8⁺ memory T cell subpopulation, T_n(-like) cells, that was described by the authors in a previous publication.

4. The authors made it clear that skewed interpretation of scRNA-seq data due to low event counts was mitigated by using aggregate data from multiple donors.

5. In depth analysis of scRNA-seq and Cite-seq datasets was already raised by reviewer 1 (point #6) and addressed by the authors.

We thank the reviewer for considering whether the concerns previously raised by reviewer 2 have been addressed. We appreciate the reviewer's assessment that the majority of these points have now been resolved. In response to point 2, and to provide additional context, we previously cited two published studies and have now added one additional reference demonstrating that naïve-like memory T cells (T_{NM}) are induced by yellow fever and smallpox vaccination (Fuertes Marraco et al. 2015, Adamo et al. 2023, Akondy et al. 2017). T_{NM} cells differ from *bona fide* naïve T (T_N) cells: they exhibit enhanced recall capacity upon re-stimulation, including greater expansion, higher expression of activation markers, and increased IFN- γ release. In addition, T_{NM} cells are maintained at higher frequencies decades after vaccination compared to truly naïve cells in unvaccinated individuals. This provides a clearer background on T_{NM} biology and their functional relevance within memory T-cell subsets. We clarified that these subsets have been previously described, while keeping the introduction concise to preserve clarity in the main text.

A few specific things:

1. The authors mentioned a few times in the manuscript (e.g. Figure 4) and rebuttal, that they interrogated metabolic flux in antigen-specific T cells. However, SCENITH experiments do not examine 'metabolic flux' in cells, it assesses metabolic dependencies of protein translation. All mention of 'metabolic flux' in this context should be removed.

We agree that SCENITH does not directly measure metabolic flux. Therefore, throughout the manuscript, we removed the term “metabolic flux” when referring to SCENITH data to avoid incorrect interpretation and instead used the term “metabolic analysis through protein translation dynamics” (**Fig. 2**) or “metabolic pathway dependence of protein translation” (**Fig. 4**). Of note, we also clearly described in the abstract as well as at several instances in the main text that we measured protein translation as a surrogate readout for metabolic activity.

*2. Although this may be considered a matter of semantics, I find the use of the term $T_n(-like)$ to describe the quiescent memory T cell subset problematic. The inclusion of brackets around “-like” is unnecessary and potentially confusing. A more conventional nomenclature, such as T_n -like, would be clearer and less ambiguous. The current usage could be misinterpreted to suggest that the identified subset contains both naïve and memory T cells. This ambiguity is further compounded in Figures S11–S13, where the experiments appear to analyze *bona fide* naïve cells (CD62L+CD45RA+) rather than naïve-like memory cells, and yet the $T_n(-like)$ term was still used.*

We appreciate the reviewer's suggestion regarding terminology. We understand that the “ $T_N(-like)$ ” notation may be confusing, and we have now replaced this terminology with “naïve-like memory T cells (T_{NM})” throughout the manuscript when referring to reactive or epitope-specific T cells following immunization. Otherwise, we use the well-known abbreviation “ T_N ” for antigen-inexperienced cells. In addition, for the polyclonal stimulation experiments (**Extended Data Fig. 5-7**; previously Fig. S11–S13), we now explicitly state that these cells predominantly should represent *bona fide* naïve T cells, to avoid any possible ambiguity. Following the

reviewer's suggestion, we also adapted the nomenclature to the biological context in other parts of the manuscript to be more precise. For example, we refer to "T_N/T_{NM}" Leiden clusters since these entail both antigen-experienced as well as antigen-inexperienced cells. However, we refer to A2/NS4B⁺ cells found in these Leiden clusters after immunization solely as T_{NM} cells.

3. Reviewer 2 expressed uncertainty regarding the principal finding of this manuscript. This confusion may, in part, stem from the current title. The metabolic characterization of T_n(-like) cells represents the central finding of the study and should be explicitly reflected in the title. Referring to "metabolic quiescence" without contextualizing it within the framework of T_n(-like) cell biology is potentially misleading, as bulk T_{scm}, T_{cm}, and even T_{em} populations also exhibit metabolic quiescence at later stages following antigenic challenge.

We thank the reviewer for this comment. To better emphasize the connection between metabolic quiescence and naïve-like memory T cells, we revised the title from the original broader wording to:

"Metabolic quiescence of naïve-like memory T cells precedes and maintains antigen-specific T-cell memory".

This change highlights the central finding of the manuscript while preserving the concept of metabolic quiescence.